# Calibration of an airborne $HO_X$ instrument using the All Pressure Altitude based Calibrator for $HO_X$ Experimentation (APACHE)

Daniel Marno[1], Cheryl Ernest[1*], Korbinian Hens[1**], Umar Javed[1,2], Thomas Klimach[1], Monica Martinez[1], Markus Rudolf[1], Jos Lelieveld[1], and Hartwig Harder[1]

[1] Atmospheric Chemistry Department, Max Planck Institute for Chemistry, 55128, Mainz, Germany
[2] Forschungszentrum Jülich GmbH, IEK-8, 52425, Jülich, Germany
*now at: Department of Neurology, University Medical Center of the Johannes Gutenberg University Mainz, 55131, Mainz, Germany
**now at: Hübner GmbH & Co KG – Division Hübner Photonics, 34123 Kassel, Germany

Correspondence: Daniel Marno (daniel.marno@mpic.de), Hartwig Harder (hartwig.harder@mpic.de)

**Abstract.** Laser induced fluorescence (LIF) is a widely used technique for both laboratory-based and ambient atmospheric chemistry measurements. However, LIF instruments require calibrations in order to translate instrument response into concentrations of chemical species. Calibration of LIF instruments measuring OH and $HO_2$ ($HO_X$), typically involves the photolysis of water vapor by 184.9 nm light thereby producing quantitative amounts of OH and $HO_2$. For ground-based $HO_X$ instruments, this method of calibration is done at one pressure (typically ambient pressure) at the instrument inlet. However, airborne $HO_X$ instruments can experience varying cell pressures, internal residence times, temperatures, and humidity during flight. Therefore, replication of such variances when calibrating in the lab are essential to acquire the appropriate sensitivities. This requirement resulted in the development of the APACHE (All Pressure Altitude-based Calibrator for $HO_X$ Experimentation) chamber, to characterize the sensitivity of the airborne LIF-FAGE $HO_X$ instrument, HORUS, which took part in an intensive airborne campaign, OMO-ASIA 2015. It utilizes photolysis of water vapor, but has the additional ability to alter the pressure at the nozzle of the HORUS instrument. With APACHE, the HORUS instrument sensitivity towards OH (26.1 - 7.8 cts $s^{-1}$ $pptv^{-1}$ $mW^{-1}$, $\pm$ 22.6% 1$\sigma$) and $HO_2$ (21.2 - 8.1 cts $s^{-1}$ $pptv^{-1}$ $mW^{-1}$, $\pm$ 22.1% 1$\sigma$) was characterized to the external pressure range at the instrument nozzle of 227 - 900 mbar. Measurements supported by a computational fluid dynamics model, COMSOL multiphysics, revealed that, for all pressures explored in this study, APACHE is capable of initializing homogenous flow and maintaining near uniform flow speeds across the internal cross-section of the chamber. This reduces the uncertainty regarding average exposure times across the mercury (Hg) UV ring lamp. Two different actinometrical approaches characterized the APACHE UV ring lamp flux as 6.37 x $10^{14}$ ($\pm$ 1.3 x $10^{14}$) photons $cm^{-2}$ $s^{-1}$. One approach used the HORUS instrument as a transfer standard in conjunction with a calibrated on-ground calibration system traceable to NIST standards, which characterized the UV ring lamp flux to be 6.9 ($\pm$ 1.1) x$10^{14}$ photons $cm^{-2}$ $s^{-1}$. The second approach involved measuring ozone production by the UV ring lamp using an ANSYCO O3 41 M ozone monitor, which characterized the UV ring lamp flux to be 6.11 ($\pm$ 0.8) x$10^{14}$ photons $cm^{-2}$ $s^{-1}$. Data presented in this study are the first direct calibrations of an airborne $HO_X$ instrument, performed in a controlled environment in the lab using APACHE.

## 1    Introduction

It is well known that the hydroxyl (OH) radical is a potent oxidizing agent in daytime photochemical degradation of pollutants sourced from anthropogenic and biogenic processes thus accelerating their removal from our atmosphere. The hydroperoxyl radical ($HO_2$) also

plays a central role in atmospheric oxidation as it not only acts as a reservoir for OH, but is involved in formation of other oxidants such as peroxides and impacts the cycling of pollutants such as $NO_X$ (= NO + $NO_2$) (Lelieveld et al., 2002). Therefore, measurements of OH and $HO_2$ ($HO_X$) within the troposphere are essential in understanding the potential global scale impacts of pollutants in both the present day and in climate predictions. One common $HO_X$ measurement method is Laser Induced Fluorescence (LIF) (Stevens et al., 1994; Brune et al., 1995; Hard et al., 1995; Martinez et al., 2003; Faloona et al., 2004; Stone et al., 2010; Hens et al., 2014; Novelli et al., 2014). Other methods have been successfully implemented to measure $HO_X$. Chemical Ionization Mass Spectrometry (CIMS) (Cantrell et al., 2003; Mauldin et al., 2004; Sjostedt et al., 2007; Dusanter et al., 2008; Kukui et al., 2008; Albrecht et al., 2019) and Differential Optical Absorption Spectroscopy (DOAS) (Brauers et al., 1996; Brauers et al., 2001; Schlosser et al., 2007) have also been used in the measurement of $HO_X$ in the field and in intercomparison projects with LIF instrumentation. However, low atmospheric concentrations of $HO_X$ (Schlosser et al., 2009) and potential interferences (Faloona et al., 2004; Fuchs et al., 2011; Mao et al., 2012; Hens et al., 2014; Novelli et al., 2014; Fuchs et al., 2016) can make $HO_X$ measurements especially challenging. Airborne LIF-FAGE (LIF-Fluorescence Assay by Gas Expansion) instruments experience large variability in pressure, humidity, instrument internal air density, and internal quenching during flights, which cause a wide array of instrumental sensitivities (Faloona et al., 2004; Martinez et al., 2010; Regelin et al., 2013; Winiberg et al., 2015). Therefore, it is critical to utilize a calibration system that can suitably reproduce in-flight conditions to determine the instrument response to known levels of OH and $HO_2$ to acquire robust $HO_X$ measurements.

The first stage of the Hydroxyl Radical measurement Unit based on fluorescence Spectroscopy (HORUS) inlet is an inlet pre-injector (IPI), used to determine the concentration of background OH interferences by removing atmospheric OH from the signal via addition of an OH scavenger such as propane. IPI draws 50-230 sL $min^{-1}$ depending on altitude and is susceptible to temperature and pressure-driven changes in internal reaction rates and residence times under flight conditions. This has implications for the removal of atmospheric OH in the inlet and for the characterization of background interference signals in HORUS. Therefore, a device capable of providing stable high flows whilst reproducing a wide range of pressures and temperatures is needed in order to calibrate the airborne HORUS instrument. This led to the production, characterization, and utilization of the calibration device APACHE (All Pressure Altitude based Calibrator for $HO_X$ Experimentation) which is described in depth in this work.

## 2      Experimental design and set up

### 2.1   APACHE design overview

Figure 1 shows the overview of the APACHE system. In front of the APACHE inlet, a series of mixing blocks are installed where multiple dry synthetic air additions are injected into a controlled humidified air supply ensuring thorough mixing of water vapor before being measured by a LI-COR 6262 $CO_2$/$H_2O$ (Figure 1a). This air is then fed into a large mass flow controller (MFC). The construction of the APACHE chamber itself is shown in Figure 1b. The first section contains the diffuser inlet with a sintered filter (bronze alloy, Amtag, filter class 10). This 2 mm thick sintered filter, with a pore size of 35 $\mu$m, initializes a homogeneous flow and further improves the mixing of water vapor in front of the UV ring lamp (described further in section 4). The water photolysis section contains a low-pressure, 0.8 A, mercury ring lamp (uv-technik, see supplementary, Figure S.1) which produces a constant radial photon flux at

184.9 nm, situated 133 mm after the sintered filter and separated from the main APACHE chamber by an airtight quartz window. Between the lamp and the quartz window there is an anodized aluminum band with thirty 8 mm apertures blocking all light apart from that going through the apertures, which reduces the amount of UV flux entering APACHE and limits the

**(a)** APACHE Injection scheme

**(b)** APACHE overview

Legend:

**SF:** Sintered Filter          **P$_{SP}$:** Perforated stainless steel plates with wool mesh.

**UVL:** UV Lamp housing          **HI:** HORUS Inlet (IPI)          **PT:** Pitot tube

**Figure 1.** Overview of the APACHE system and the pre-mixing set up used in the lab to calibrate the HORUS airborne instrument. A picture at the bottom shows the perforated stainless steel plates with wool mesh.

size of the illuminated area. The IPI system is clamped down 169.5 mm behind the photolysis section in such a way that the instrument sample flow is perpendicular to the airflow passing over the IPI nozzle. The nozzle protrudes 51.5 mm into the APACHE cavity much like it is when installed in the aircraft shroud system (see Figure 2), and is made air tight with the use

of O-rings. Opposite the IPI nozzle, there is an airtight block attachment containing a series of
monitoring systems. A pitot tube attached to an Airflow PTSX-K 0-10Pa differential pressure
sensor (accuracy rating of 1% at full scale, 1σ) is used to monitor the internal flow speeds
within APACHE. A 3 kOhm NTC-EC95302V thermistor is used to monitor the air temperature
and an Edwards ASG2-1000 pressure sensor (with an accuracy rating of ± 4 mbar, 2σ) monitors
the static air pressure. Additionally, there are two one-quarter inch airtight apertures in the
monitoring block that can be opened to enable other instrumentation to be installed.

## 2.2 Pressure control

For this study, the operational pressure range of APACHE used was 227 – 900 mbar, with
precision of ± 0.1% (1σ) and accuracy of ± 2% (1σ) with mass flows ranging from 200 to 990
sL min$^{-1}$. This was achieved using an Edwards GSX160 scroll pump controlling the volume
flow in combination with a MFC (Bronkhorst F-601A1-PAD-03-V) controlling the mass flow
of air entering APACHE. This system reached air speeds of 0.9 to 1.5 m s$^{-1}$ through APACHE
at pressures ranging from 250 to 900 mbar and at temperatures ranging from 282 to 302 K.
Temperature changes inside APACHE are not controlled. However, as air temperature is
measured throughout the calibration device and HORUS, any term that is affected by
temperature is characterized using the corresponding measured temperature values. Although
not critical for this study, the operational pressure range of APACHE can be extended by
changing the draw speed of the Edwards scroll pump. However, that may cause the flow speeds
and potentially the flow speed profiles across the UV ring lamp to vary in between different
pressure calibrations.

## 2.3 The airborne HORUS instrument

The LIF-FAGE instrument developed by our group (HORUS), is based on the original
design of GTHOS (Ground Tropospheric Hydrogen Oxide Sensor) described by Faloona et al.
(2004) and is described in further detail by Martinez et al. (2010). The airborne instrument is a
revised and altered design to perform under conditions experienced during flight and conform
to aeronautical regulations. It was primarily developed for installation on the High Altitude and
Long Range Research Aircraft (HALO) and took place in the OMO-Asia 2015 airborne
campaign. The system comprises of an external inlet shroud, detections axes, laser system, and
a vacuum system (Figure. 2). Additionally, this is the first airborne LIF-FAGE instrument
measuring HO$_X$ with a dedicated inlet pre injector (IPI) system installed for the purpose of
removing atmospheric OH enabling real time measurements and quantification of potential
chemical background OH interferences, OH-CHEM (Mao et al., 2012). The airborne IPI
system is redesigned to fit within the shroud inlet system and its walls are heated to 30 °C,
whilst maintaining similar operational features as the on-ground IPI installation (Novelli et al.,
2014). To prevent excessive collisions of OH and HO$_2$ with the IPI nozzle and internal walls,
thus limiting losses of HO$_X$ during flight, the momentum inertia of the air passing through the
external shroud system had to be overcome to promote flow direction into the instrument. This
was achieved by installing a choke point behind the IPI nozzle in the inlet shroud, resulting in
a reduction in air flow speed. For example without the shroud choke, flow speeds in excess of
200 m s$^{-1}$ could occur in the shroud during flight. However, with the choke point, flow speeds
in the shroud during flight did not exceed 21 m s$^{-1}$ during OMO-Asia 2015, which is sufficiently
below the sample velocities of IPI during flight (44 – 53 m s$^{-1}$). Additionally, it limits non-
parallel flows across the IPI nozzle created by variable pitch, roll and yaw changes of the
aircraft. As the aircraft changes pitch, roll and yaw, the measured OH variability increases by
± 4.51 x10$^4$ cm$^{-3}$ (1σ), which is only 10 to 15 % higher than the natural variability of OH. This
increase in variability is negligible as is represents, depending on internal pressure, 19 to 30 %
of the detection limit of the instrument. Both these effects of the external shroud improve the
measurement performance by reducing variable wall losses of $HO_X$ at the IPI nozzle under
flight conditions. The IPI system (with a nozzle orifice diameter of 6.5 mm) samples (51 to 230
sL min$^{-1}$) from the central air flow moving through the internal shroud. A critical orifice is
located at the end of IPI in the center of the IPI cross section, which enables the HORUS
instrument to sample (3 to 17 sL min$^{-1}$) from the central flow moving through IPI. This further
reduces influences of wall loss within IPI on the overall measured signal in the cells. The
removal of excess flow moving through IPI occurs via a perforated ring that surrounds the base
of the critical orifice cone, evacuated by a blower.

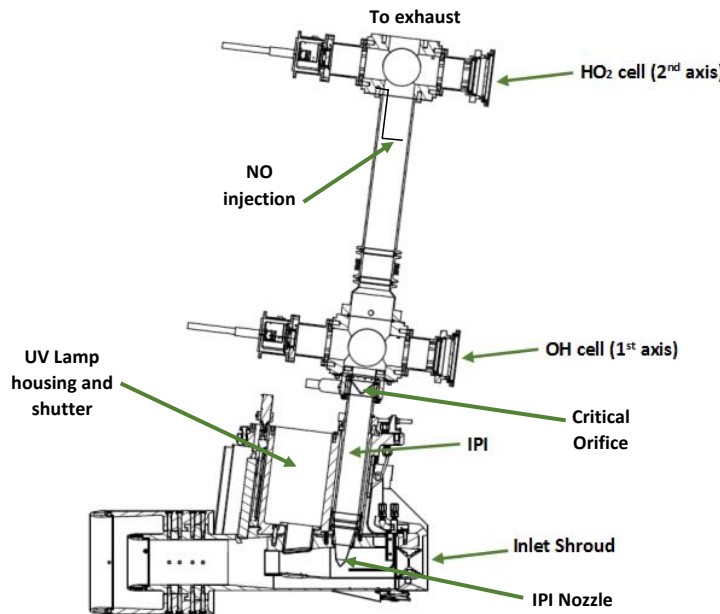

**Figure 2.** Overview of the airborne HORUS system as installed in the HALO aircraft. $HO_2$ is measured indirectly through the addition of NO that quantitatively converts $HO_2$ into OH. The NO injection occurs via a stainless steel 1/8 inch line, shaped into a ring perpendicular to the airflow with several unidirectional apertures of 0.25 mm diameter creating essentially a NO shower.


As with other LIF-FAGE $HO_X$ instruments, HORUS measures an off-resonance signal to
discern the net OH fluorescence signal. This is achieved by successive cycling of the laser
tuning from on-resonance (measuring the total signal of OH fluorescence and the signal
originating from other fluorescence and electronic sources), to off-resonance (measuring all
the above except the OH fluorescence). The HORUS instrument utilizes the $Q_1(2)$ transition
$X^2\Pi_{3/2}(v'' = 0) \rightarrow A^2\Sigma^+(v' = 0)$ (Freeman. 1958; Dieke and Crosswhite. 1962; Langhoff et
al., 1982; Dorn et al., 1995; Holland et al., 1995; Mather et al., 1997). The net OH signal ($S_{OH}$)
is the difference between the on-resonance and off-resonance signals, OH-WAVE (Mao et al.,
2012). The OH sensitivity ($C_{OH}$) and average laser power within the detection axis ($Wz_{1\ pwr}$)
are then used to calculate the absolute OH mixing ratio (see Eq. (1)). $HO_2$ is measured indirectly

through the quantitative conversion of atmospheric $HO_2$ to OH by injection of nitric oxide (NO) under the low-pressure conditions within HORUS.

$$HO_2 + NO \rightarrow NO_2 + OH \tag{R1}$$

When NO is injected into the instrument, both ambient OH and $HO_2$ are measured in the second detection axis. The net $HO_2$ signal ($S_{HO2}$) in the second axis is therefore derived from subtracting the net OH signal from the first detection axis normalized by the ratio of the OH sensitivities for the two detection axes ($C_{OH(2)} / C_{OH}$) from the net $HO_X$ signal ($S_{HOx}$). Then $S_{HO2}$ is corrected by the sensitivity to $HO_2$ ($C_{HO2}$) and laser power ($Wz_{2\,pwr}$) to reach absolute $HO_2$ mixing ratio (see Eq. (2)).

$$[OH] = \frac{S_{OH}}{(C_{OH} \cdot Wz_{1\,pwr})} \tag{1}$$

$$[HO_2] = \frac{1}{(C_{HO_2} \cdot Wz_{2\,pwr})} \cdot \left\{ S_{HO_X} - \frac{(C_{OH(2)} \cdot Wz_{2\,pwr})}{(C_{OH} \cdot Wz_{1\,pwr})} S_{OH} \right\} \tag{2}$$

where, $Wz_{1\,pwr}$ is the laser power in the first detection axis, $Wz_{2\,pwr}$ is the laser power in the second detection axis and $C_{OH}$ and $C_{HO2}$ are the calibrated sensitivity factors for OH and $HO_2$ (cts $s^{-1}$ $pptv^{-1}$ $mW^{-1}$) respectively. By calibrating using a known OH mixing ratio, the instrument sensitivity $C_{OH}$ can be determined by rearranging Eq. (1) to:

$$C_{OH\,(cal)} = {S_{OH_{cal}}} \Big/ {([OH] \cdot Wz_{1\,pwr})} \tag{3}$$

The sensitivity of HORUS depends on the internal pressure, water vapor mixing ratios, and temperature, which are subject to change quite significantly during flight. Therefore, further parameterization when calibrating is required to fully constrain the sensitivity response of the instrument at various flight conditions. Eq. (4) shows the terms that affect the sensitivity of the first HORUS axis that measures OH.

$$C_{OH}(P,T) = c0 \cdot \rho_{Int}(P,T) \cdot Q_{IF}(P,T,H_2O) \cdot b_c(T) \cdot [\alpha_{IPI}(P,T) \cdot \alpha_{HORUS}(P,T)] \tag{4}$$

where c0 is determined by calibrations and is the lump sum coefficient of all the pressure independent factors affecting the HORUS sensitivity, for example, OH absorption cross section at 308nm, the photon collection efficiency of the optical setup and quantum yield of the detectors, as well as pressure independent wall loss effects. For calibrations, c0 is normalized by laser power and has the units (cts $pptv^{-1}$ $s^{-2}$ $cm^3$ $molecule^{-1}$ $mW^{-1}$). $\rho_{Int}$ is the internal molecular density. $Q_{IF}$ is the quenching effect (s), which consists of the natural decay frequency of OH, OH decay due to collisional quenching that is dependent on pressure, temperature, and water vapor mixing ratio, and the detector opening and closing gating times after the initial excitation laser pulse. Both are pressure dependent terms as denoted in Eq. (4). The Boltzmann correction ($b_c$) has a temperature dependency as it corrects for any OH molecules that enter the HORUS instrument in a thermally excited state and are therefore not measurable by fluorescence excitation at the wavelength used. $\alpha$ is the pressure dependent OH transmission, which is the fraction of OH that reaches the point of detection. This term is separated for the two-tier pressure conditions present in the instrument. The term $\alpha_{IPI}$ represents the correction for pressure and temperature dependent OH loss on the walls within IPI. The term $\alpha_{HORUS}$ is the correction for pressure dependent OH loss to the walls within the HORUS detection axes post critical orifice. Whilst the quenching effects, internal densities and Boltzmann corrections

can be quantified by calculation, and the power entering the measurement cell is measured, the
two factors that need to be determined through calibration are c0 and OH transmission, α. Once
the c0 coefficient and α terms are known, the final in-flight measured OH mixing ratio (pptv)
is found:
$$[OH] = \left.S_{OH}\middle/(c0 \cdot \rho_{Int} \cdot Q_{IF} \cdot b_c \cdot [\alpha_{IPI} \cdot \alpha_{HORUS}] \cdot Wz_{1\,pwr})\right. \tag{5}$$
As $S_{OH}$ scales with laser power, the terms that describe the instrument sensitivity shown as
the denominator in Eq. (5), which ultimately have the units cts s$^{-1}$ pptv$^{-1}$ mW$^{-1}$, must also be
scaled to the measured laser power ($Wz_{1\,pwr}$) during flight to acquire the absolute measurement
of OH mixing ratio. As depicted in both Figure 1b and Figure 2, the complete system is
calibrated with IPI attached and operating as it did when installed in the aircraft. Therefore, the
combined losses of OH within IPI and in the low pressure regime post critical orifice (that has
a diameter of 1.4 mm) contribute to the overall calibrated $C_{OH}$ sensitivity factor in the same
way during measurement and calibrations, meaning that the OH transmission of HORUS can
be quantified with both OH transmission terms ($\alpha_{IPI}$ and $\alpha_{HORUS}$) combined into one term
($\alpha_{Total}$).
$$[OH] = \left.S_{OH}\middle/(c0 \cdot \rho_{Int} \cdot Q_{IF} \cdot b_c \cdot [\alpha_{Total}] \cdot Wz_{1\,pwr})\right. \tag{6}$$
Figure 3 shows the schematic of the different factors described above and their impact on
the overall sensitivity.

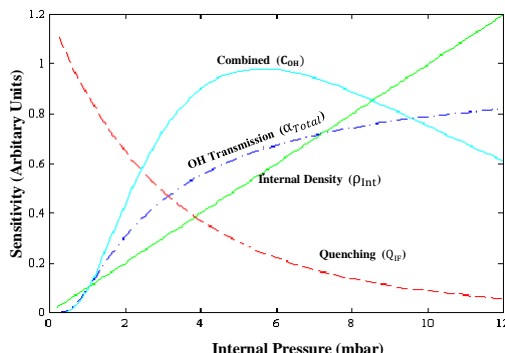

**Figure 3.** A schematic showing the overall sensitivity curve as a function of internal pressure (light blue line), OH transmission (dotted-dashed dark blue line), internal density (green line) , and the quenching (dashed red line).

**3        Calibration method and theory**
As an overview, Table 1 shows common calibration techniques for OH instruments. The
APACHE system is based on the production of known quantified and equal concentrations of
OH and HO$_2$ via photolysis of water vapor in only synthetic air using a Hg ring lamp emitting
UV radiation at 184.9 nm.
$$H_2O + h\upsilon \xrightarrow{\lambda=184.9\ nm} OH + H^* \tag{R2}$$
$$H^* + O_2 \xrightarrow{O_2} OH + O_3 \tag{R3}$$
$H^* \xrightarrow{M} H$ (R4)
$H + O_2 \xrightarrow{M} HO_2$ (R5)
**Table 1.** Various known methods for OH instrument calibrations

| | Technique | Method | Quoted (1σ) Uncertainty | Limitations | References |
|---|---|---|---|---|---|
| (I) | Water UV-Photolysis | See sections 3 and 4 | 10-30% | Dependent on lamp, photon flux measurement, and absorption | (Creasey et al., 2003; Heard and Pilling. 2003; Holland et al., 2003; Ren et al., 2003; Faloona et al., 2004; Smith et al., 2006; Martinez et al., 2010; Mallik et al., 2018) |
| (II) | Pulsed $N_2$-$H_2O$ RF discharge | At low pressure (0.1 Torr); OH and NO produced using a low power RF discharge. Concentrations of NO and OH are closely linked | 20% | Requires NO measurement using stable ambient air calibrations | (Dilecce et al., 2004; Verreycken and Bruggeman. 2014) |
| (III) | Low-pressure flow-tube RF discharge | OH radical production by titration of H atoms with $NO_2$. Known amount of H atoms produced using microwave discharge using low pressure flow tube | 30% | Stable ambient air calibrations | (Stevens et al., 1994) |
| (IV) | Continuously Stirred Tank Reactor and decay of select hydrocarbons | In a CSTR, OH produced through UV-irradiation of humidified air flow with injection of a specific Hydrocarbon (1,3,5-trimethylbenzene, $C_9H_{12}$) and NO. More recent studies have used Cyclohexane, n-pentane and iso-butene Concentrations of OH relates to decay rate of the Hydrocarbon | 24-36% | Time intensive, systematic wall loss of OH in reactor | (Hard et al., 1995; Hard et al., 2002; Winiberg et al., 2015) |
| (V) | Steady-State $O_3$-alkene | A steady state OH concentration produced from ozonolysis of a known concentration of an alkene | 42% | Time consuming, large uncertainties compared to other methods | (Heard and Pilling. 2003; Dusanter et al., 2008) |
| (VI) | Laser photolysis of Ozone | Photolysis of $O_3$ with 284 nm light producing $O(^1D)$. Which then reacts with $H_2O$ producing OH | 40-50% | Requires large apparatus | (Tanner and Eisele. 1995) |


Stable water mixing ratios with a variability of $< 2\%$ were achieved by heating 300 sL min$^-$
$^1$ flow of synthetic air to 353 K and introducing deionized water using a peristaltic pump into
this heated gas flow causing it to evaporate before entering a 15 L mixing chamber. This
prevents re-condensation and humidity spikes when the pump is introducing the water. The
humidified gas flow is then diluted (to around 3 mmol $mol^{-1}$) and mixed further with additional
dry pure synthetic air via a series of mixing blocks to achieve the required and desired stable
water vapor mixing ratios. The photolysis of $H_2O$ has only one spin-allowed and energetically
viable dissociation channel at 184.9 nm (Engel et al., 1992), meaning the quantum yield of OH
and H* are unified (Sander et al., 2003). Even though reaction R3 is possible particularly since
the H* atoms can carry transitional energies of 0.7 eV at 189.4nm (Zhang et al., 2000), the fast
removal of energy by reaction R4 allows for the general assumption that all H * atoms produced
leads to $HO_2$ production (Fuchs et al., 2011). The use of water photolysis as a OH and $HO_2$
radical source for calibration of $HO_X$ instruments has been adopted in a number of studies
(Heard and Pilling. 2003; Ren et al., 2003; Faloona et al., 2004; Dusanter et al., 2008; Novelli
et al., 2014; Mallik et al., 2018).  As an example, the factors required to quantify the known
concentrations of OH and $HO_2$ during calibrations are shown below:
$$[OH] = [HO_2] = [H_2O] \cdot \sigma_{H_2O} \cdot F_{184.9\,nm} \cdot \phi_{H_2O} \cdot \; t \hspace{4cm} (7)$$
where in Eq. (7), the OH and $HO_2$ concentrations are a product of photolysis of a known
concentration of water vapor $[H_2O]$, $\sigma_{H2O}$ is the absorption cross section of water vapor, 7.22
($\pm$ 0.22) x $10^{-20}$ $cm^2$ $molecule^{-1}$ at 184.9 nm (Hofzumahaus et al., 1997; Creasey et al., 2000).
$F_{184.9\,nm}$ is the actinic flux (photons $cm^{-2}$ $s^{-1}$) of the mercury lamp used for photolysis, $\phi_{H2O}$ is
the quantum yield and t is exposure time. The quantum yield of water vapor photolysis at the
184.9 nm band is 1 (Creasey et al., 2000).
**4       Results and Discussion**
**4.1 Flow conditions**
With any calibration device, the flow conditions must be characterized to inform subsequent
methods and calibrations. Regarding APACHE, the  two main factors to be resolved are (i)
how uniform are the flow speed profiles and therefore exposure times in respect to the
APACHE cross section, and (ii) the impact of OH wall losses.
To this end, experimental and model tests were performed to determine whether the
combination of the sintered filter, and the stainless steel perforated plates and wool
arrangement could provide a homogeneous flow. This means that under operation the flow
speeds should be uniform along the cross section of APACHE to within the uncertainty of the
measurements. This is to ensure that the air masses passing across the lamp have the same
exposure times irrespective of where they are in the cross section. Additionally, model
simulations can provide an indication of, as a function of APACHE pressure, the development
and scale of boundary air conditions where air parcels experience extended contact time with
the interior walls of APACHE, and so have pronounced OH wall losses. This highlights
potential flow conditions where there is sufficient time between the photolysis zone and the IPI
nozzle to allow APACHE boundary air to expand into and influence the OH content of the air
being sampled by HORUS.
**4.1.1 Flow speed profiles**
During calibration, the pressures within the HORUS instrument had to be controlled and
monitored to replicate the in-flight conditions. The APACHE chamber pressure is equivalent
to the in-flight pressure in the shroud where the HORUS system samples. The pressure of the
detection axes depends on the pressure at the IPI nozzle and the efficiency of the pumps. Within
IPI itself, the airflow through it is dependent on the pressure gradient between the shroud and
the ambient pressure at the IPI exhaust or alternatively the APACHE pressure and pressure in
front of the XDS 35 scroll pump (post IPI blower). During the campaign, the exhausts of all
blowers and pumps of the HORUS system were attached to the passive exhaust system of the
aircraft and were thus exposed to ambient pressure. Therefore, the same IPI blower and pumps
that were installed on HALO were used in the lab, and throughout the calibrations the pressure
at the exhaust for every blower and pumps involved in the HORUS instrument was matched to
the respective in-flight ambient pressures by attaching a separate pressure sensor, needle valve
and XDS35 scroll pump system. Additionally, to match the power that is provided on the
aircraft, a 3-phase mission power supply unit was used to power the pumps in the lab during
testing and throughout the calibrations. Figure 4 shows the lab setup described above.

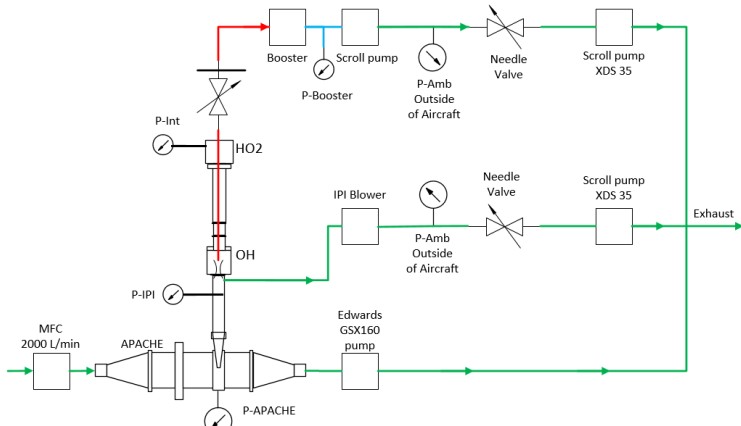

**Figure 4.** The experimental setup with the additional needle valves, pressure sensors and XDS35 scroll pumps attached to the exhausts of all pumps and blowers of HORUS to match in-flight pumping efficiencies when calibrating with APACHE. The red lines depict the low-pressure region within HORUS, the blue is the pressure monitoring line between the booster and scroll pump that drive the HORUS sample flow, and the green show the external gas lines.

To limit the effect of wall loss, HORUS samples air from the core of the APACHE flow
system and draws only a fraction of the total air flow as shown in Figure 5. At 900 hPa the
HORUS instrument takes 20 % and at 275 hPa HORUS takes 30 % of the total volume flow
entering APACHE. To validate that this proportional volume flow into HORUS does not
disturb the flow conditions within APACHE, flow speed profiles were performed using the
Prandtl pitot tube installed directly opposite the IPI nozzle, which can be positioned flush
against the internal wall up to 60.5 mm into the APACHE cavity, which is 15 mm from the
APACHE center. Figure 6 shows the measured flow speed profile (blue data points) when the
APACHE pressure was 920 hPa. As the distance between the APACHE wall and the pitot tube
inlet increased, no significant change in the flow speed was observed. The largest change
observed was between 46.6 and 60.5 mm where the flow speed increased by 0.16 m s$^{-1}$, which
is 22.8 % smaller than the combined uncertainty of these two measurements $\pm$ 0.21m s$^{-1}$ (2σ).
Compared to the other four measurement points performed at 920 mbar, the 1.54 m s$^{-1}$
measured at 60.5 mm is not significantly different. However, when performing the speed
profile tests at lower pressures, the pressure difference measured was close to or below the

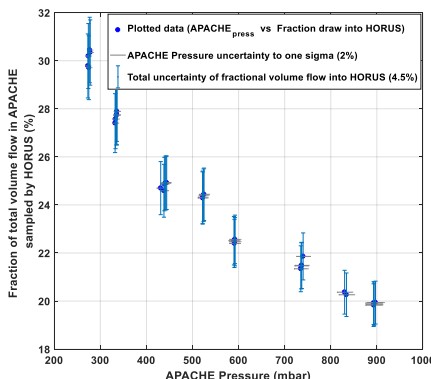

**Figure 5.** The percentage of the total volume flow entering APACHE, which is sampled by HORUS as a function of pressure within APACHE. All error bars are quoted to 1σ.

resolution of the differential pressure sensor. Consequently, the flow inside APACHE and the
IPI nozzle was simulated using the computational fluid dynamics (CFD) model from COMSOL
multiphysics to gain a better understanding of the flow speed profiles at all pressures. The CFD
module in COMSOL uses Reynolds Averaged Navier-Stokes (RANS) models (COMSOL.
2019). The standard k-epsilon turbulence model with incompressible flows was used for this
study as it is applicable when investigating flow speeds below 115 m s$^{-1}$ (COMSOL. 2019).
An extra fine gridded mesh of a perforated plate with a high solidity ($\sigma_s = 0.96$) was
implemented in the turbulence model to generate the turbulence and replicate the flows created
by the bronze sintered filter (Roach. 1987). The model was constrained with the pressures
measured within APACHE and IPI. The volume flow was calculated from the measured mass
flow entering APACHE and temperatures were constrained using the thermistor readings. To
gain confidence in the model, the flow speed output data was compared to the available
measured flow speed profile, see Figure 6.

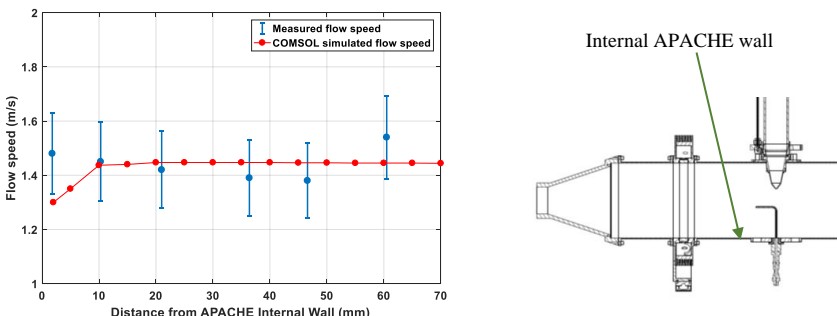

**Figure 6.** The measured (blue) and COMSOL simulated (red) flow speed profiles within APACHE, at 920 hPa. The x-axis is the distance from the internal wall of APACHE. The error bars are quoted to 2σ.


Overall, the modelled flow speed profile did not differ significantly from measured. The
only point where the model significantly disagreed with measurements was at the boundary (<
4 mm away from the APACHE wall), where the model predicted a flow speed of 1.3 m s$^{-1}$,
which is 6 % lower than the minimum extent of the measurement uncertainty 1.38 m s$^{-1}$. This
disagreement could also be due to the uncertainty in the parametrization of the boundary
conditions in the COMSOL simulations. However, as this is occurring within a region that
ultimately does not influence the air entering HORUS, see section 4.1.2, the disagreement
between modelled and measured flow speeds at distances less than 4 mm from the APACHE
wall is ignored.  Figure 7 shows the simulated flow speeds at six discrete pressures within
APACHE.

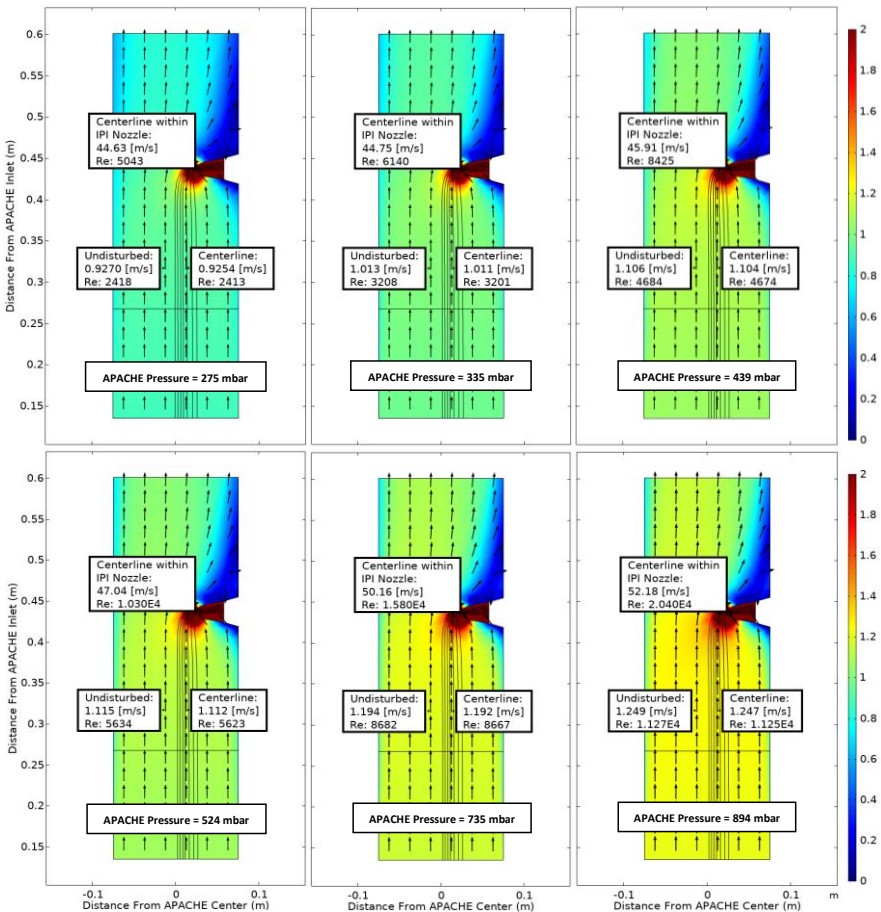

**Figure 7.** COMSOL Multiphysics output data, simulating the flow speed conditions at 6 discrete pressures within APACHE ranging from 275 to 894 mbar, between the sintered filter and the first perforated stainless steel plate. The color represents flow speed in m s$^{-1}$. The black lines are the streamlines created by the HORUS sample flow. The black arrows depict the flow direction. The x-axis is the distance from the center of APACHE in meters. The y-axis is the distance from the APACHE inlet. The "centerline within the IPI nozzle" tags show the flow conditions in the center of the fully formed flows after the HORUS pinhole, the "undisturbed" tags show the flow conditions outside of the HORUS streamlines, and the "centerline" tags show the flow conditions in the center of the streamlines (i.e. the area of flow influenced by HORUS sampling).


The black lines depict the streamlines of the HORUS sample flow and the color gradient relates
to the flow speed. The flow conditions in the center flow within the IPI nozzle, the center of
the streamlines and the undisturbed flow airflow not influenced by the sample flow of the
HORUS instrument are indicated. The Figure shows the internal APACHE dimensions starting
from the sintered filter to the first perforated stainless steel plate 0.135 m and 0.601 m from the
APACHE inlet, respectively. From the simulations, the centerline flow speed differs by less
than 0.1 % compared to the undisturbed flow, which is also the case at 275 mbar when HORUS
is drawing in the highest percentage of the total volume flow entering APACHE. After the
sintered filter the high calculated Reynolds numbers (Re > 2300) support the statement that a
turbulent flow regime is created. Additionally, the measurements in conjunction with
simulations show that the small pores of the sintered filter release a uniform distribution of
small turbulent elements across the diameter of APACHE, which remain prevalent all the way
up to the IPI nozzle.

### 354 4.1.2 HO$_X$ losses in APACHE

The modelled OH mixing ratios (pptv) in Figure 8 show the change in OH content as the air
flows along the length of APACHE. Mixing ratios were used as they are independent of the
changing density within APACHE. In every simulation, the OH and HO$_2$ concentrations were
initialized at zero, and losses at the walls were fixed to 100 % for both OH and HO$_2$. The radial
photolytic production of OH and HO$_2$, as calculated using Eq. (7) and Eq. (9), occurred when
the air passed the UV ring lamp. For all simulations, the HO$_X$ radical-radical recombination
loss reactions, (reactions R6-R8), and the measured molecular diffusion coefficient of OH$_{Dm}$
in air (Tang et al., 2014) was used:
$\text{OH}_{Dm} = 179 \ (\pm 20) \ \text{Torr cm}^2 \ \text{s}^{-1}$ $\quad\quad\quad (239 \pm 27 \ \text{hPa cm}^2 \ \text{s}^{-1})$
In literature, there have been no reports of successfully performed tests that accurately
measure HO$_2$ diffusivity coefficients in air. However, calculations of HO$_2$ diffusion
coefficients using the Lennard-Jones potential model have been performed (Ivanov et al.,
2007). Ivanov et al. (2007) performed a series of measurements and Lennard-Jones potential
model calculations to quantify the polar analogue diffusion coefficients for OH, HO$_2$ and O$_3$ in
both air and pure helium. The calculated OH and O$_3$ diffusion coefficients in air from the
Lennard-Jones potential model were in good agreement with the recommended measurement
values in Tang et al., (2014) well within the given uncertainties. Therefore, to best replicate the
diffusivity of HO$_2$ within the simulations, the following diffusion coefficient of HO$_2$ in air from
the Ivanov et al., (2007) paper was used:
$\text{HO}_{2 \ Dm} = 107.1 \ \text{Torr cm}^2 \ \text{s}^{-1}$ $\quad\quad\quad (142.8 \ \text{hPa cm}^2 \ \text{s}^{-1})$
It is clear from Figure 8, that irrespective of pressure the air masses at the boundary (where
wall losses are 100 %) do not have sufficient time to expand into the HORUS sample flow
streamlines, and influence HO$_X$ content entering HORUS. Lateral exchanges between air at the
walls of APACHE and the free air in the center are suppressed due to the preservation of the
small turbulence regime between the sintered filter and IPI. Table 2 provides, for six pressures,
the evolution of OH along the length of APACHE, within the streamlines created by the
HORUS sample flow as depicted in Figure 8.
In Table 2, the L term represents OH mixing ratios on the left-most HORUS sample flow
streamline shown in Figures 7 and 8. C represents OH mixing ratios in the center of the
HORUS sample flow streamlines shown in Figures 7 and 8. R represents OH mixing ratios on
the right-most HORUS sample flow streamline shown in Figures 7 and 8. The mean mixing
ratio at each APACHE pressure does not change significantly and is thus independent of the
distance from the lamp. Conversely, the standard deviations of the OH mixing ratios within the
HORUS sampling streamlines decrease as the distance from the lamp increases, indicating that
the air is homogenizing. However, Figure 8 and Table 2, with support from available
measurements, indicate that the OH-depleted air masses (i.e. air masses that have experienced
loss of OH on the APACHE walls) do not expand into and influence the OH content of air that
is being sampled by HORUS. The main loss process that influences $HO_X$ entering HORUS is
the wall loss occurring at the IPI nozzle itself. According to the COMSOL simulations, around
22.2 ($\pm$ 0.8) % (1$\sigma$) of OH and $HO_2$ is lost at the nozzle. This value does not significantly
change with pressure, indicating that the $HO_X$ loss at the nozzle is pressure independent. As
described in section 2.3, the pressure independent sensitivity coefficients are a lump sum value
containing the pressure independent wall losses for OH and $HO_2$. Therefore, the characterized
pressure independent sensitivity coefficients, shown in section 4.3, have the OH and $HO_2$ losses
at the IPI nozzle constrained within them.

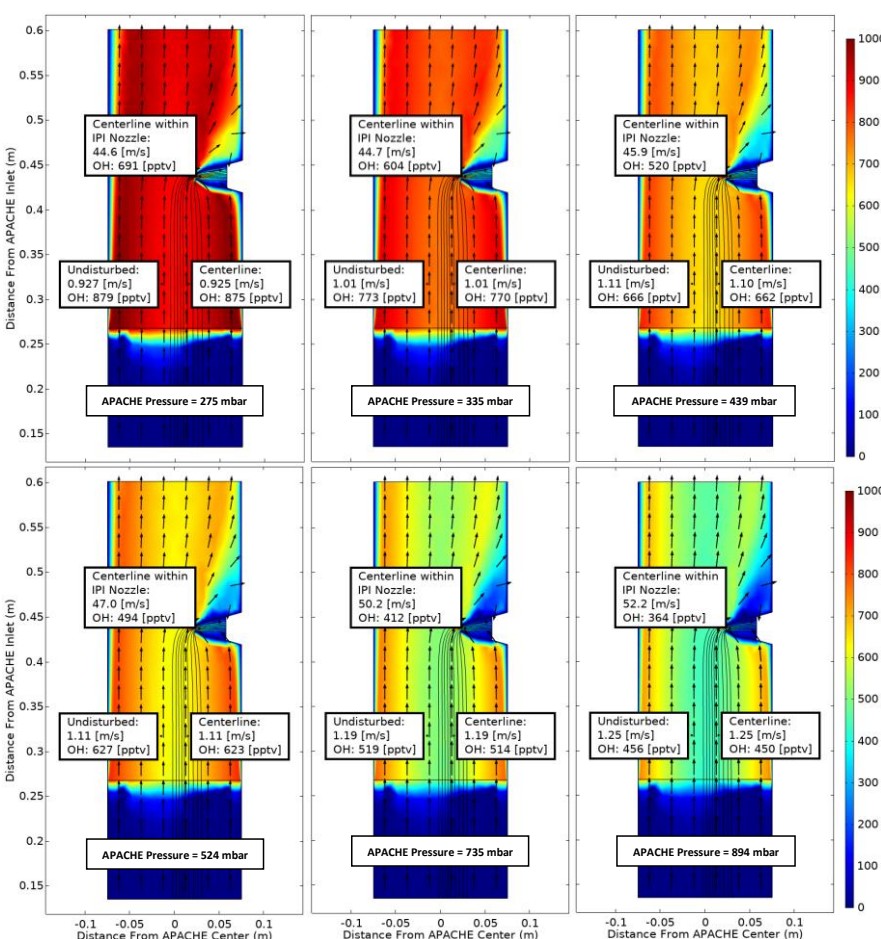

**Figure 8.** COMSOL Multiphysics output data, simulating OH conditions at 6 discrete pressures within APACHE ranging from 275 to 894 mbar, between the sintered filter and the first perforated stainless steel plate. The color is OH mixing ratio (pptv), with initial OH production occurring at the lamp (0.26 m from APACHE inlet), using Eq. (7) and Eq. (9), with water vapour mixing ratios kept constant at 3.2 mmol mol$^{-1}$. The black lines are the streamlines created by the HORUS sample flow. The black arrows depict the flow direction. The x-axis is the distance from the center of APACHE in meters. The y-axis is the distance from the APACHE inlet. The "centerline within IPI nozzle" tags represent the flow and OH concentrations in the center of the fully formed flows after the HORUS pinhole. The "undisturbed" tags show the flow conditions outside of the HORUS streamlines, and the "centerline" tags show the flow conditions in the center of the streamlines (i.e. influenced by HORUS sampling).


**Table 2.** The evolution of OH within the HORUS sample flow streamlines, along the length of APACHE at all six pressures, within the streamlines created by HORUS sampling as depicted in figure 8. The L term represents OH mixing ratios on the left most streamline, C represents OH mixing ratios in the center of the streamlines, and R represents OH mixing ratios on the right most streamline. The centerline within IPI nozzle column shows the OH mixing ratios in the center of the flow in the HORUS inlet. All standard deviations are quoted to 1σ.

| APACHE Pressure (mbar) | OH (pptv) At the lamp | | | | | OH (pptv) 4.2 cm from lamp | | | | | OH (pptv) 8.4 cm from lamp | | | | | OH (pptv) 12.8 cm from lamp | | | | | OH (pptv) 2 cm before HORUS Inlet | | | | | In Centerline within IPI Nozzle (pptv) |
|---|---|---|---|---|---|---|---|---|---|---|---|---|---|---|---|---|---|---|---|---|---|---|---|---|---|---|
| | L | C | R | Mean | Std (1σ) | L | C | R | Mean | Std (1σ) | L | C | R | Mean | Std (1σ) | L | C | R | Mean | Std (1σ) | L | C | R | Mean | Std (1σ) | |
| **894** | 438 | 445 | 513 | 465 | 41.4 | 442 | 446 | 507 | 465 | 36.4 | 438 | 455 | 500 | 464 | 32.0 | 442 | 456 | 501 | 466 | 30.8 | 445 | 457 | 490 | 464 | 23.3 | 364 |
| **735** | 502 | 508 | 572 | 527 | 38.8 | 506 | 509 | 567 | 527 | 34.4 | 502 | 519 | 560 | 527 | 29.8 | 507 | 519 | 562 | 529 | 28.9 | 509 | 521 | 550 | 527 | 21.1 | 412 |
| **524** | 611 | 617 | 672 | 633 | 33.6 | 615 | 619 | 668 | 634 | 29.5 | 613 | 627 | 660 | 633 | 24.1 | 617 | 628 | 664 | 636 | 24.6 | 619 | 629 | 651 | 633 | 16.4 | 493 |
| **439** | 652 | 657 | 706 | 672 | 29.8 | 656 | 659 | 702 | 672 | 25.7 | 654 | 666 | 698 | 673 | 22.7 | 657 | 667 | 699 | 674 | 21.9 | 660 | 669 | 686 | 672 | 13.2 | 520 |
| **335** | 760 | 765 | 805 | 777 | 24.7 | 764 | 766 | 801 | 777 | 20.8 | 762 | 773 | 799 | 778 | 19.0 | 766 | 774 | 803 | 781 | 19.5 | 768 | 776 | 788 | 777 | 10.1 | 603 |
| **275** | 866 | 871 | 907 | 881 | 22.4 | 870 | 872 | 907 | 883 | 20.8 | 869 | 879 | 904 | 884 | 18.0 | 873 | 880 | 905 | 886 | 16.8 | 875 | 882 | 889 | 882 | 7.0 | 689 |


**4.2 UV conditions**

The photolysis lamp is housed in a mount with the side facing into the chamber having an anodized aluminum band with thirty 8 mm apertures installed between the lamp and a quartz wall. The housing was flushed with pure nitrogen to purge any $O_2$ present before the lamp was turned on. The nitrogen flushing was kept on continuously thereafter. After approximately one hour, the lamp reached stable operation conditions, i.e the relative flux emitted by the lamp as measured by a photometer (seen in Figure 1b at the UVL on the underside of the APACHE chamber) was constant. The flux ($F_\beta$) entering APACHE is not the same as the flux experienced by the molecules sampled by HORUS ($F$). Factors influencing the ratio between $F_\beta$ and $F$ are as follows. (i) Absorption of light by $O_2$, which is particularly important as $O_2$ has a strong absorption band at 184.9 nm and the $O_2$ density changes in APACHE when calibrating at the different pressures. (ii) The variable radial flux, which is dependent on the geometric setup of the ring lamp and on the location within the irradiation cross section where the molecule is passing. These factors were resolved through the combination of two actinometrical crosscheck methods. The advantage of actinometrical methods is that the flux calculated is derived directly from the actual flux that is experienced by the molecules themselves as they pass through the APACHE chamber.

The first actinometrical method (A) used the HORUS instrument as a transfer standard to relate the flux of a pre-calibrated penray lamp used on the ground based calibration device to $F_\beta$ entering APACHE. This entailed first calibrating the HORUS instrument using a pre-characterized ground based calibration device (Martinez et al., 2010). The pre-calibrated penray lamp flux ($\phi_0$) is calculated from the measured NO concentrations that are produced by irradiating a known mixture of $N_2O$ in a carrier gas:

$$\phi_0 = \frac{(k_a\,[N_2][M] + k_b[N_2] + k_c[N_2O] + k_d[N_2O][NO])}{2k_d[N_2O]^2\sigma_{N2O}f_{N2O}} \tag{8}$$

where $\sigma_{N2O}$ is the absorption cross section of $N_2O$ at 184.9 nm and $f_{N2O}$ is the correction factor that accounts for the flux reduction via absorption by $N_2O$. A TEI NO monitor measures the NO concentration. For more details on how the ground calibration device is characterized using the photolysis of N2O in conjunction with a TEI NO monitor, see Martinez et al. (2010). Since the pre-characterized ground based calibration device is designed to supply only 50 sL min$^{-1}$, and the sensitivity of airborne HORUS instrument is optimized for high altitude flying, the critical orifice diameter in HORUS was changed from the airborne configuration of 1.4 mm to a 0.8 mm on-ground* configuration. Additionally, the IPI system was switched to passive (i.e. the exhaust line from IPI to the IPI blower was capped). This was to adapt HORUS to a mass flow that the ground based calibration device is able to provide and reduces the internal pressure within HORUS (from 18 mbar to 3.5 mbar) to optimize the sensitivity towards OH at ambient ground level pressures (~1000 mbar). The asterisk discerns terms that were quantified when the smaller 0.8 mm critical orifice was used. The calculated instrument on-ground* sensitivity was then used to translate OH and $HO_2$ concentrations produced by the uv-technik Hg ring lamp into a value for $F_\beta$. Take note that for the direct calibrations of the airborne HORUS system using the characterized APACHE system, discussed in section 4.3, the same initial 1.4 mm diameter critical orifice as used during the airborne campaign was installed. The HORUS on-ground* sensitivities at 1010 mbar for OH and $HO_2$ are 13.7 ($\pm$ 1.9) cts s$^{-1}$ pptv$^{-1}$ mW$^{-1}$ and 17.9 ($\pm$ 2.5) cts s$^{-1}$ pptv$^{-1}$ mW$^{-1}$ respectively, with the uncertainties quoted to 1$\sigma$. This sensitivity was then used to calculate the OH and $HO_2$ concentrations at the instrument

nozzle with the APACHE system installed and operating at 1010 mbar. To ensure sufficient
flow stability during calibration at this high pressure, the Edwards GSX160 scroll pump was
disengaged. Additionally, the water mixing ratios were kept constant (~3.1 mmol mol$^{-1}$) and
oxygen levels were varied by adding different pure $N_2$ and synthetic air mixtures, via MFCs.
The OH and $HO_2$ concentrations at the IPI nozzle were 1.41 ($\pm$ 0.01) and 1.31 ($\pm$ 0.01) x 10$^{10}$
molecules cm$^{-3}$ respectively when using a water vapor mixing ratio of 3.1 mmol mol$^{-1}$ in
synthetic air injected into APACHE. The uncertainties are quoted as measurement variability
at 1$\sigma$. Using these values, the OH and $HO_2$ concentrations at the lamp were back calculated
accounting for radical-radical loss reactions (R6-R8) and $HO_X$ reactions with $O_3$ (R9-R10)
using rate constants taken from Burkholder et al. (2015) with temperature (T) in Kelvin.
$OH + OH \rightarrow H_2O + O(^3P)$        $k = 1.8 \times 10^{-12} \cdot \exp^{\left[\frac{1}{T}\right]}$        (R6)
$HO_2 + HO_2 \rightarrow H_2O_2 + O_2$       $k = 3.0 \times 10^{-13} \cdot \exp^{\left[\frac{460}{T}\right]}$        (R7)
$OH + HO_2 \rightarrow H_2O + O_2$        $k = 4.8 \times 10^{-11} \cdot \exp^{\left[\frac{250}{T}\right]}$        (R8)
$HO_2 + O_3 \rightarrow OH + 2\,O_2$        $k = 1.0 \times 10^{-14} \cdot \exp^{\left[\frac{-490}{T}\right]}$      (R9)
$OH + O_3 \rightarrow HO_2 + O_2$        $k = 1.7 \times 10^{-12} \cdot \exp^{\left[\frac{-940}{T}\right]}$     (R10)
In APACHE when the Edwards GSX160 scroll pump was disengaged, the transit time
between the UV radiation zone and the IPI nozzle was 0.18 seconds, resulting in chemical
losses of 30 to 33 % for OH and 27 to 30 % $HO_2$, depending on oxygen concentration.
Accounting for these chemical losses yields, OH concentrations of 2.0 ($\pm$ 0.02) x 10$^{10}$
molecules cm$^{-3}$ and $HO_2$ concentrations of 1.9 ($\pm$ 0.02) x 10$^{10}$ molecules cm$^{-3}$ at the lamp, at
1010 mbar. The photon flux ($F$) experienced by the air sampled by HORUS, quantified using
the OH and $HO_2$ concentrations stated above, ranged from 3.8 x10$^{14}$ photons cm$^{-2}$ s$^{-1}$ to 6.7
x10$^{14}$ photons cm$^{-2}$ s$^{-1}$ depending on oxygen concentrations and considering the chemical
losses. As described before, Eq. (7) shows how the production of OH at the lamp is calculated:
$[OH] = [H_2O] \cdot \sigma_{H_2O} \cdot F_{184.9\,nm} \cdot \phi_{H_2O} \cdot t$         (7)
$F_{184.9\,nm}$ is the actinic flux encountered by the water molecules as they pass across the
photolysis region, which is dependent on the attenuation of the flux ($F_\beta$) entering APACHE
due to water vapor and $O_2$ molecules. Whereas the absorption coefficient of water vapor is
constant across the linewidth of the 184.9 nm Hg emission line, the effective absorption cross
section of molecular oxygen ($\sigma_{O2}$) changes significantly at 184.9 nm within the linewidth of
the Hg lamp (Creasey et al., 2000). Therefore, $\sigma_{O2}$ affecting the APACHE calibrations is
dependent on $O_2$ concentration, and the ring lamp temperature and current. Since the operating
temperature of the uv-technik Hg lamp and the current applied (0.8 A) was kept constant during
the actinometrical experiments and during the APACHE calibrations, any effect on $\sigma_{O2}$
regarding the ring lamp linewidth does not need to be investigated further in this study. The
relationship of $F_{184.9\,nm}$ to $F_\beta$ can be derived using Beer-Lambert principles:
$F_{184.9\,nm} = F_\beta \cdot e^{-\left(\gamma_{H_2O}[H_2O] + \gamma_{O_2}[O_2]\right)}$        (9)
where $F_\beta$ is the flux intensity entering APACHE from ring lamp, with:
$$\gamma_{O2} = R_\beta \cdot \omega \cdot \sigma_{O2} \qquad (10)$$
where $R_\beta$ is the radial distance of the sampled air parcel to the ring lamp of APACHE, $\omega$ a
correction factor replicating the integrated product of the absorption cross section and the ring
lamp's emission line as modified by the effect of the absorption of $O_2$ present in between the
lamp and the flight path of the sampled air, normalized by $\sigma_{O2}$ is the effective cross section of
$O_2$. When combining Eq. (7) and Eq. (9) the OH concentration produced at the lamp is
quantified as:
$$[OH] = [H_2O] \cdot \sigma_{H_2O} \cdot \phi_{H_2O} \cdot t \cdot F_\beta \cdot e^{-(\gamma_{H_2O}[H_2O] + \gamma_{O_2}[O_2])} \qquad (11)$$
Eq. (11) can be rearranged to:
$$\ln\left[\frac{[OH]}{[H_2O]}\right] = \ln(F_\beta \cdot t \cdot \phi_{H_2O} \cdot \sigma_{H_2O}) + (-\gamma_{H_2O} \cdot [H_2O] - \gamma_{O_2} \cdot [O_2]) \qquad (12)$$
Figure 9, shows the measured production of OH, (left side of Eq. (12)) plotted against
oxygen concentration. Given that the other terms within Eq. (12) are constant with changing
oxygen levels, the plotted gradient of the linear regression in Figure 9 yields $\gamma_{O2}$ as a function
of oxygen concentration being $1.2 \times 10^{-19}$ ($\pm 0.05 \times 10^{-19}$) cm$^3$ molecule$^{-1}$.

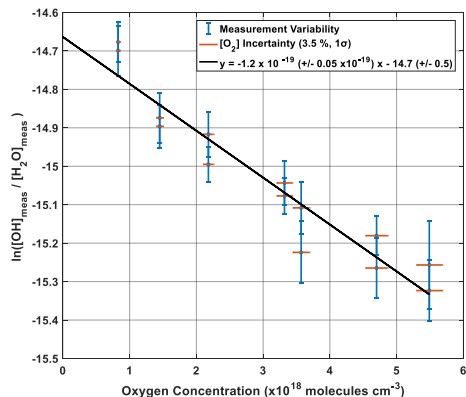

**Figure 9.** Plot showing the result of Eq. (11) as a function of oxygen concentration.


Given that, the y intercept of the linear regression, -14.66, is equal to the natural logarithm
of ($F_\beta$ t $\phi_{H2O}$) minus ($\gamma_{H2O}$ [$H_2O$]), the flux entering APACHE $F_\beta$ can be characterized:
$$F_\beta = \left(\frac{e^{-14.66}}{t \cdot \phi_{H2O}}\right) - (\gamma_{H_2O} \cdot [H_2O]) = 6.9 \times 10^{14} (\pm 1.1 \times 10^{14}) \text{ photons cm}^{-2} \text{ s}^{-1} \qquad (13)$$
The accuracy in $F_\beta$ from method A is 15.9 % (1$\sigma$). Table 3 shows the parameters and their
uncertainties contributing to the $F_\beta$ characterized in method A.




**Table 3.** Parameters and uncertainties used in method A, using HORUS as a transfer standard. Overall uncertainty
is the sum of the quadrature of the individual uncertainties. $O(^1D)$ yield is taken from Martinez et al., (2010).

| Parameter | Comments | Total Uncertainty (1σ) |
|---|---|---|
| NO Monitor (TEI) | Calibration uncertainty | 5.2 % |
| NO standard (NPL) | Purity and concentration of the gas | 1 % |
| $N_2O$ cross section | JPL recommendation | 2 % |
| $H_2O$ cross section | JPL recommendation | 2 % |
| $\gamma_{O2}$ | From method A | 3.5 % |
| $O(^1D)$ yield | Martinez et al. (2010) | 1 % |
| Kinetic rate coefficients | JPL recommendation | 12 % |
| $F_\beta$ Variability | From method A | 3.5 % |
| Photolysis chamber dimensions | Specifications of in-house workshop | 3 % |
| $[H_2O]$ | Calibration with NIST standard  Dew point generator | 2 % |
| $[O_2]$ | From method A | 3.4 % |
| Mass flow controllers | Calibration with NIST DryCal | 2 % |
| Pressure and Temperature sensors | Validated against NIST standard | 2 % |
| Overall Experimental Stability | Variability of measured terms | 4 % |
| Overall uncertainty | | 15.9 % |


The second actinometrical method (B) involved using an ANSYCO O3 41 M ozone monitor
to measure the ozone mixing ratio profile between the IPI nozzle and the wall surface of
APACHE, at ground pressure (1021 mbar). This method utilizes $O_2$ photolysis at 184.9 nm,
which produces two $O(^3P)$ atoms capable of reacting with a further two $O_2$ molecules to
produce $O_3$.

$$O_2 + h\nu \rightarrow O(^3P) + O(^3P) \tag{R11}$$

$$O_2 + O(^3P) + M \rightarrow O_3 + M \tag{R12}$$

The value of $1.2 \times 10^{-19}$ cm$^3$ molecule$^{-1}$ for $\gamma_{O2}$ found in the previous method was used to
calculate the actinic flux entering APACHE:

$$F_\beta = \frac{[O_3]}{[O_2] \cdot \gamma_{O_2} \cdot \phi_{O_2} \cdot t \cdot e^{-(\gamma_{O2}[O_2])}} \tag{14}$$

$\Phi_{O2}$ is the quantum yield of $O_2$ at 184.9 nm, which has been determined to be 1 between
242 and 175 nm (Atkinson et al., 2004). As in method A, the ozone produced at the lamp is
quantified by back calculating from the ozone measured at the ANSYCO O3 41 M inlet
position. Inside APACHE, typical ozone concentrations ranged from $1.26 \times 10^{12}$ to $2.05 \times 10^{12}$
molecules cm$^{-3}$ depending on the oxygen concentration. From this approach, the calculated $F_\beta$
is $6.11 \times 10^{14}$ ($\pm 0.8 \times 10^{14}$) photons cm$^{-2}$ s$^{-1}$ with a total uncertainty of 12.9 % (1σ). The final
value taken for $F_\beta$ is the average of the two experiments, weighted by their uncertainties:

Actinic flux $(F_\beta) = 6.37 \times 10^{14}$ ($\pm 1.3 \times 10^{14}$) photons cm$^{-2}$ s$^{-1}$

Accuracy in $F_\beta = 20.5$ % (1σ)

Agreement for $F_\beta$ between method A and B, Zeta score = 0.59.

Table 4 shows the parameters and their uncertainties which contribute to the $F_\beta$ characterized in method B.

**Table 4.** Parameters and uncertainties involved in Method B, using ANSYCO O3 41 M monitor. The total uncertainty is the sum of the quadrature of the individual uncertainties.

| Parameter | Comments | Total Uncertainty (1σ) |
|---|---|---|
| O$_3$ calibrator | Calibrated against a primary standard | 2 % |
| [O$_3$] | Calibration of ANSYCO O3 41 M monitor | 4 % |
| [O$_2$] | From method A | 3.4 % |
| $\gamma_{O2}$ | From method A | 3.5 % |
| F$_\beta$ Variability | From method A | 3.5 % |
| Mass flow controllers | Calibration with NIST DryCal | 2 % |
| Pressure and Temperature sensors | Validated against NIST standard | 2 % |
| Experimental Stability | Variability of values | 10.1 % |
| Overall uncertainty | | 12.9 % |

## 4.3 Evaluation of instrumental sensitivity

Figure 10 shows the sensitivity curve of HORUS, the quenching effect, the linear fits used to quantify the pressure independent sensitivity coefficients, and relative HO$_X$ transmission values for OH , OH in the second axis, and HO$_2$ plotted as a function of the HORUS internal density. The red smoothed line in Figure 10 row A represents the calculated sensitivity curve for each measurement using Eq. (4) and the characterized variables therein. Given that this calculated sensitivity curve for each measurement agrees to within 2 sigma of the uncertainties in measured calibration curves, we are confident that each of the terms described in Eq. (4) have been sufficiently resolved. Table 5 shows the ranges, precision and uncertainties of measured or calculated variables affecting OH and HO$_2$ concentrations formed in APACHE.

**Table 5.** Parameters within APACHE, their ranges and uncertainties, contributing to the uncertainty in the three measurement sensitivities within HORUS.

| Parameter (unit) | Range or typical value | Precision (1σ) | Total Uncertainty (1σ) |
|---|---|---|---|
| $F_\beta$ at 184.9 nm (photons cm$^{-2}$ s$^{-1}$) | 6.37 x 10$^{14}$ | 3.5 % | 20.5 % |
| $\sigma$H$_2$O (cm$^2$ molecule$^{-1}$) | 7.22 x 10$^{-20}$ | - | 2 % |
| $\gamma_{O2}$ (cm$^3$ molecule$^{-1}$) | 1.22 x 10$^{-19}$ | 1.8 % | 3.5 % |
| [O$_2$] (x10$^{18}$ molecules cm$^{-3}$) | 1.1 - 4.8 | 1.4 % | 3.4 % |
| [H$_2$O] (x10$^{16}$ molecules cm$^{-3}$) | 2.00 - 7.41 | 1.2 % | 2 % |
| Mass flow controller (sL min$^{-1}$) | 203 - 988 | < 2 % | 2 % |
| Pressure sensors (mbar) | 275 - 900 | < 1 % | 2 % |
| Temperature sensors (K) | 282 - 302 | < 1 % | 2 % |
| Overall | | 5 % | 21.5 % |

The pressure independent sensitivity coefficients (cN) for OH in the 1$^{st}$ axis (c0), OH in the 2$^{nd}$ axis (c1), and HO$_2$ in the 2$^{nd}$ axis (c2), are calculated by rearranging Eq. (4) to:

$$c0 \; \cdot \; \rho_{Int}\,(P,T) \; = \frac{C_{OH}(P,T)}{Q_{IF}(P,T) \cdot b_c(T) \cdot [\alpha_{IPI\,OH}\,(P,T) \cdot \alpha_{HORUS\,OH}(P,T)]} \tag{15}$$

$$c1 \cdot \rho_{Int}(P,T) = \frac{C_{OH(2)}(P,T)}{Q_{IF(2)}(P,T) \cdot b_c(T) \cdot [\alpha_{IPI\,OH}(P,T) \cdot \alpha_{HORUS\,OH(2)}(P,T)]}$$ (16)
$$c2 \cdot \rho_{Int}(P,T) = \frac{C_{HO2}(P,T)}{Q_{IF(2)}(P,T) \cdot b_c(T) \cdot [\alpha_{IPI\,HO2}(P,T) \cdot \alpha_{HORUS\,HO2}(P,T)]}$$ (17)
The products of Eq. (15 to 17) are plotted against internal density in Figure 10 row C, where
the slopes of the linear regressions are the pressure independent sensitivity coefficients. Note
that in Eq. (16) and Eq. (17), the bracketed 2 terms are in relation to the OH measurement at
the second axis. Table 6 shows the values, precision and uncertainty in the quantified pressure
independent sensitivity coefficients.
**Table 6.** Pressure independent sensitivities and their overall uncertainty from calibrations with APACHE.

| Parameter (cts pptv$^{-1}$ s$^{-2}$ cm$^3$ molecule$^{-1}$ mW$^{-1}$) | Value (x10$^{-9}$) | Precision ($\pm 1\sigma$) | Total Uncertainty ($1\sigma$) |
|---|---|---|---|
| c0 for OH in OH axis | 3.8 | 4 % | 6.9 % |
| c1 for OH in HO$_2$ axis | 2.3 | 4 % | 6.9 % |
| c2 for HO$_2$ in HO$_2$ axis | 4.5 | 2 % | 5.6 % |

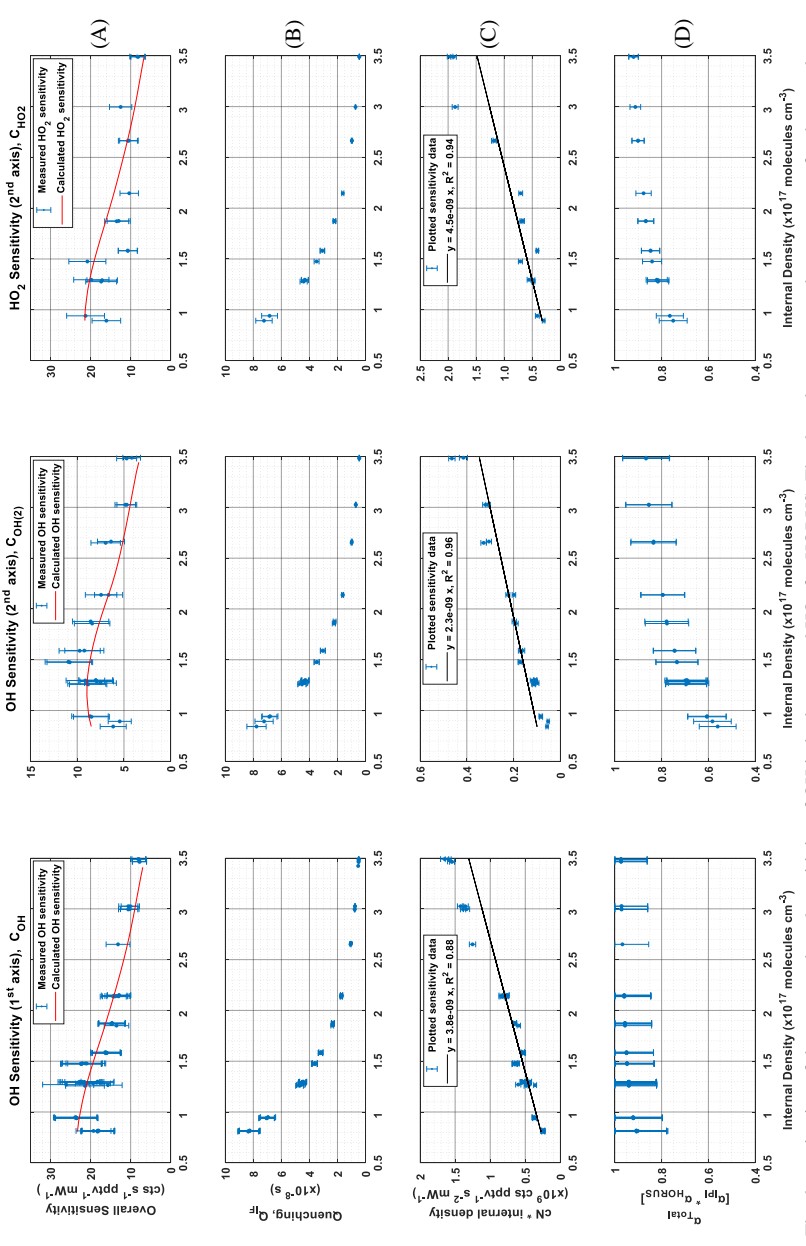

**Figure 10.** The determination of the pressure based sensitivity of OH in both axes and HO$_2$ for HORUS. The data shown are one-hour averages for the tested pressures, all plotted with the internal density on the x-axis. The top row (A) is the measured (blue data points) HORUS sensitivity curve and calculated (red line) sensitivity curve. The second row (B) is internal quenching by N$_2$, O$_2$, and water vapor, and row (C) is internal density and cN (c0 for OH 1$^{st}$ axis, c1 for OH 2$^{nd}$ axis and c2 for HO$_2$), (D) is the total OH and HO$_2$ transmissions, all plotted against internal density. The error bars represent measurement variability (1σ), for rows B and C. In rows A and D, the error bars represent the total uncertainty (1σ).

In Figure 10, row C the quenching ($Q_{IF}$) is plotted against internal density. $Q_{IF}$ is calculated
using the same approach as described in Faloona et al. (2004) and Martinez et al. (2010):

$$Q_{IF}(P) = \frac{1}{\Gamma}(e^{-\Gamma g_1} - e^{-\Gamma g_2}) \tag{18}$$

where $\Gamma$ is the excited state decay frequency (Hz), consisting of the natural decay frequency,
and decay due to collisional quenching that is dependent on pressure, temperature, and water
vapor mixing ratio. $g_1$ and $g_2$ are the detector gate opening and closing times after the initial
excitation laser pulse, which are set to 104 ns and 600 ns respectively.
As described in section 2.3, the pressure independent sensitivity coefficients are lump sum
variables containing pressure independent $HO_X$ wall loss. The pressure dependent $HO_X$
transmission through the HORUS instrument is quantified and described below. In-flight, IPI
operates across the pressure range of 180 to 1010 mbar. However, within HORUS, post critical
orifice, at detection axes where $HO_X$ is measured the pressure ranges from 3.1 to 18.4 mbar.
Therefore, the transmission through IPI ($\alpha_{IPI}$) and through HORUS ($\alpha_{HORUS}$) must be quantified
separately using the corresponding measured pressures and transit times, before being
combined as the total transmission ($\alpha_{IPI} \cdot \alpha_{HORUS} = \alpha_{Total}$). To calculate the transmission of $HO_X$
within IPI, the following was used:

$$\alpha_{IPI\,OH} = 1 - \left\lceil \frac{OH_{DM}(P) \cdot t_{r\,IPI}\,(P,T) \cdot \pi}{IPI_A \cdot P_{IPI}} \right\rceil \tag{19}$$

$$\alpha_{IPI\,HO_2} = 1 - \left\lceil \frac{HO_{2\,DM}(P) \cdot t_{r\,IPI}\,(P,T) \cdot \pi}{IPI_A \cdot P_{IPI}} \right\rceil \tag{20}$$

where $t_{rIPI}$ is the transit time within IPI, i.e. the time it takes for air to flow from the IPI
nozzle to the critical orifice of HORUS. $IPI_A$ is the internal cross sectional area of IPI and $P_{IPI}$
is the measured pressure within IPI. The $OH_{DM}$ and $HO_{2\,DM}$ terms are the OH and $HO_2$ diffusion
coefficients as described in section 4.1.2. $\alpha_{IPI\,OH}$ is the transmission of OH through IPI, and
$\alpha_{IPI\,HO2}$ is the transmission of $HO_2$ through IPI. By applying Eq. (19) and Eq. (20), $\alpha_{IPI\,OH}$ and
$\alpha_{IPI\,HO2}$ ranged from 0.97 to 0.99 and 0.99 to 0.997 respectively across the pressure range
within IPI of 198 – 808 mbar and IPI transit times of 90 – 120 milliseconds. However, to
calculate $\alpha_{Total}$, the OH and $HO_2$ transmission post critical orifice, $\alpha_{HORUS\,OH}$ and $\alpha_{HORUS\,HO2}$,
must be resolved. $\alpha_{HORUS}$ regarding OH and $HO_2$ can be calculated by adapting Eq. (19) and
Eq. (20) to the internal HORUS conditions producing:

$$\alpha_{HORUS\,OH} = 1 - \left\lceil \frac{OH_{DM}(P) \cdot t_{r1}\,(P,T) \cdot \pi}{HORUS_A \cdot P_{int}} \right\rceil \tag{21}$$

$$\alpha_{HORUS\,OH(2)} = 1 - \left\lceil \frac{OH_{DM}(P) \cdot t_{r2}\,(P,T) \cdot \pi}{HORUS_A \cdot P_{int}} \right\rceil \tag{22}$$

$$\alpha_{HORUS\,HO_2} = 1 - \left\lceil \frac{HO_{2\,DM}(P) \cdot t_{r2}\,(P,T) \cdot \pi}{HORUS_A \cdot P_{int}} \right\rceil \tag{23}$$

where $t_{r1}$ and $t_{r2}$ are the transit times within HORUS from the critical orifice to the 1st and
2nd detection axis respectively. $HORUS_A$ is the internal cross sectional area of HORUS and $P_{int}$
is the measured internal pressure within HORUS. The OH transmission from the critical orifice
to the 1st detection cell ($\alpha_{HORUS\,OH}$) ranged from 0.93 to 0.98, the OH transmission from the
critical orifice to the 2nd detection cell ($\alpha_{HORUS\,OH\_2}$) ranged from 0.58 to 0.87, and the $HO_2$
transmission from the critical orifice to the 2nd detection cell ($\alpha_{HORUS\,HO2}$) ranged from 0.76 to
0.92. These ranges are quoted under the HORUS internal pressure range of 3.7 to 13.7 mbar

and internal transit times to the $1^{st}$ detection axis (3.8 to 4.3 milliseconds) and $2^{nd}$ detection axis (23.5 to 27.8 milliseconds). The combined $\alpha_{Total}$ values for OH, OH at the second detection axis, and $HO_2$ are plotted in Figure 10 row D as a function of the internal density of HORUS. Table 7 shows the calculated $\alpha_{Total}$ transmission terms, their precision and uncertainty for OH to the first axis, OH to the second axis, and $HO_2$ to the second axis.

**Table 7.** Pressure dependent OH and $HO_2$ transmission and their overall uncertainty from calibrations with APACHE.

| Parameter (%) | Value | Precision ($\pm 1\sigma$) | Total Uncertainty ($1\sigma$) |
|---|---|---|---|
| $\alpha_{Total}$ (for OH to OH axis) | 90 - 97 | 2.8 % | 14.3 – 11.5 % |
| $\alpha_{Total}$ (for OH to $HO_2$ axis) | 56 - 86 | 4.3 % | 14.1 – 11.5 % |
| $\alpha_{Total}$ (for $HO_2$ to $HO_2$ axis) | 75 - 92 | 2.9 % | 7.9 – 2.2 % |

Table 8 shows the measured sensitivity values using APACHE for OH at the first axis ($C_{OH}$), OH at the second axis ($C_{OH (2)}$), and $HO_2$ at the second axis ($C_{HO2}$). The precision denotes the $1\sigma$ variability in the measured $HO_X$ signals from HORUS, the total uncertainty is the root sum square of the total uncertainty values from the variables listed in Tables 5 and 6.

**Table 8.** Pressure dependent sensitivities for the three measurement within HORUS, their overall uncertainty from calibrations with APACHE. The range in the precision relates to the numbers quoted in the value column.

| Parameter (unit) | Value | Precision ($\pm 1\sigma$) | Total Uncertainty ($1\sigma$) |
|---|---|---|---|
| $C_{OH}$ (cts s$^{-1}$ pptv$^{-1}$ mW$^{-1}$) | 7.8 - 26.1 | 1.1 - 0.5 % | 22.6 % |
| $C_{OH (2)}$ (cts s$^{-1}$ pptv$^{-1}$ mW$^{-1}$) | 4.2 – 11.0 | 2.0 - 0.3 % | 22.6 % |
| $C_{HO2}$ (cts s$^{-1}$ pptv$^{-1}$ mW$^{-1}$) | 8.1 – 21.2 | 0.4 - 0.7 % | 22.2 % |

The undescribed remaining fraction that influences the instrument sensitivity ($R_{undescribed}$), is calculated by dividing the overall sensitivity values by described in Eq. (4):

$$R_{OH} = \frac{C_{OH}}{c0 \cdot \rho_{Int}(P,T) \cdot Q_{IF}(P,T) \cdot b_c(T) \cdot [\alpha_{IPI\,OH}(P,T) \cdot \alpha_{HORUS\,OH}(P,T)]} \tag{24}$$

$$R_{OH\,(2)} = \frac{C_{OH(2)}}{c1 \cdot \rho_{Int}(P,T) \cdot Q_{IF\,(2)}(P,T) \cdot b_c(T) \cdot [\alpha_{IPI\,OH}(P,T) \cdot \alpha_{HORUS\,OH(2)}(P,T)]} \tag{25}$$

$$R_{HO_2} = \frac{C_{HO2}}{c2 \cdot \rho_{Int}(P,T) \cdot Q_{IF\,(2)}(P,T) \cdot b_c(T) \cdot [\alpha_{IPI\,HO2}(P,T) \cdot \alpha_{HORUS\,HO2}(P,T)]} \tag{26}$$

$$R_{undescribed} = \{R_{OH}\,;\,R_{OH\,(2)}\,;\,R_{HO_2}\}$$

where $R_{undescribed}$ is a matrix containing the undescribed remain factors from the three measurements. When plotting $R_{undescribed}$ against the internal pressure of HORUS, (see supplementary, Figure S10), the data scatters $\pm 0.15$ ($1\sigma$) about the average value of 1.02 ($\pm 0.23$, $1\sigma$ calibration uncertainty). This means that (as an upper limit) <2 % of the overall instrument sensitivity is unresolved by the terms described in Eq. (4). Additionally, the $1\sigma$ variability of the $R_{undescribed}$ is 34 % smaller than the uncertainty in the APACHE calibration, meaning that this remaining fraction is declared as neither pressure dependent nor pressure independent.

It is important to note here that all data shown in Figure 10, with the exception of the pressure independent sensitivity coefficients, are in relation to temperatures and pressures HORUS experienced during calibrations in the lab. To apply these findings to real airborne measurements, the pressure and temperature dependent terms in Eq. (4) are calculated using the temperatures and pressures that are measured within the instrument during flight. The only terms that affect measurement sensitivity and are directly transferable from the calibrations with APACHE to the measurements in-flight shown in Eq. (4) are the pressure independent sensitivity coefficient as they are not subject to change with the large temperature and pressures ranges HORUS experiences when airborne. Figure 11 shows the pressure and temperature dependent terms from Eq. (4) characterized for a typical flight that took place during the OMO-ASIA 2015 airborne campaign. In Figure 11, the sensitivity values, limit of detection, transmission values for OH (blue data points) and $HO_2$ (red data points), and the ambient water mixing ratios (black date points) that occurred during flight 23  are plotted as a function of altitude. During flight, the OH sensitivity ranged from 5.4 ($\pm$ 1.2) cts s$^{-1}$ pptv$^{-1}$ mW$^{-1}$ on ground to 24.1 ($\pm$ 5.4) cts s$^{-1}$ pptv$^{-1}$ mW$^{-1}$ at 14 km. The HORUS sensitivity values for $HO_2$ ranged from 5.5 ($\pm$ 1.2) cts s$^{-1}$ pptv$^{-1}$ mW$^{-1}$ and reached an average maxima of 20.5 ($\pm$ 4.5) cts s$^{-1}$ pptv$^{-1}$ mW$^{-1}$ at 11.4 km. Above 11.4 km the $HO_2$ sensitivity decreased with altitude  reaching 19.7 ($\pm$ 4.4) cts s$^{-1}$ pptv$^{-1}$ mW$^{-1}$ at 14 km. This drop in $HO_2$ sensitivity is attributable to the increasing decline in $HO_2$ transmission inside HORUS as the aircraft flies higher, despite the sensitivity improvements via quenching as the air is becoming drier. The water vapor mixing ratios at 14 km on average are three orders of magnitude lower than the average water vapor mixing ratio of 1.5 % at ground level; which greatly suppresses quenching of OH and thus is the main driver for the general increasing trend in the instrument sensitivity towards $HO_X$ as altitude increases. Additionally, Figure 11 shows that the limit of detection for both OH and $HO_2$ decrease with increasing altitude. For OH, the HORUS limit of detection is ~0.11 pptv at ground level and drops to ~0.02 pptv at 14 km. For $HO_2$ the limit of detection is ~1.2 pptv at ground level and drops to 0.23 pptv at 14 km.













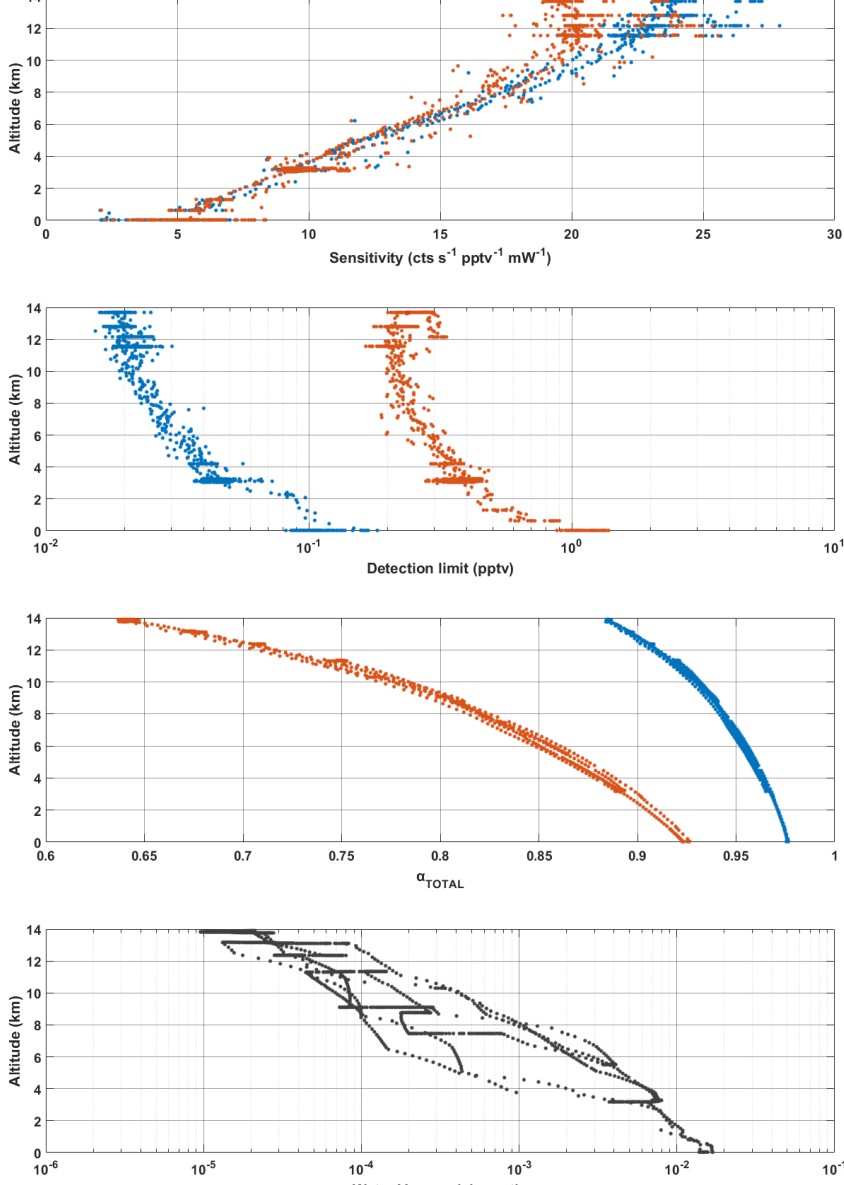

**Figure 11.** In-flight, sensitivity curves, limit of detection, and HO$_X$ transmission plotted against altitude for OH (blue data points) and HO$_2$ (red data points, and the water vapour mixing ratio (black data points) plotted against altitude in km (bottom plot). Data taken from flight 23.


## 5        Conclusions

The overall goal of this study was to develop and test a new calibration system capable of providing the high flows required by the airborne HORUS system whilst maintaining stable pressures across the pressure ranges experienced during flight. Such systems are critical to suitably characterize airborne systems, (such as a LIF-FAGE measuring $HO_X$), that have a strong pressure dependent sensitivity. In addition, this system is purely based on the use of water-vapor photolysis, which is a frequently adopted technique for $HO_X$ instrument calibration (Martinez et al., 2003; Faloona et al., 2004; Dusanter et al., 2008). The COMSOL multiphysics simulations constrained by temperature, pressure and mass flow measurements demonstrated that air masses at the boundary of the APACHE system do not have sufficient time to expand into the streamlines created by the HORUS sample flow and influence the $HO_X$ content entering HORUS. The largest uncertainties result from constraining the flux ($F_\beta$) entering APACHE ($6.37 \pm 1.3 \times 10^{14}$ photons cm$^{-2}$ s$^{-1}$ , 1σ) and the total uncertainty in the pressure independent sensitivity coefficients (ranging from 5.6 to 6.9 %, 1σ). The two actinometrical methods used to derive $F_\beta$ proved to be in good agreement with a zeta score of 0.59, considering 1σ of their uncertainties. The HORUS transfer standard method yielded an $F_\beta$ value of $6.9 \pm 1.1 \times 10^{14}$ photons cm$^{-2}$ s$^{-1}$ (1σ) and the ozone monitor method yielded an $F_\beta$ value of $6.11 \pm 0.8 \times 10^{14}$ photons cm$^{-2}$ s$^{-1}$ (1σ). Furthermore, the APACHE system enabled the total OH and $HO_2$ pressure dependent transmission factors to be characterized as a function of internal pressure. Calculations of $HO_X$ diffusivity to the walls within IPI and the low-pressure regime within HORUS yielded 90 - 97 % for OH transmission to the first detection axis, 56 - 86 % for OH transmission to the second detection axis, and 75 - 92 % for $HO_2$ transmission to the second detection axis. Future studies with APACHE are planned to expand upon the findings within this paper with a particular focus on temperature control and on improving operational pressure and flow speed ranges. However, in this study, the APACHE calibration system has demonstrated that, within the lab, it is sufficiently capable of calibrating the airborne HORUS instrument across the pressure ranges the instrument had experienced in-flight during the OMO-ASIA 2015 airborne campaign. The overall uncertainty of 22.1 – 22.6 % (1σ) demonstrates that this calibration approach with APACHE compares well with other calibration methods described earlier in Table 1. Nevertheless, there is potential for improvement. Accurate calibrations of instruments, particularly airborne instruments that have strong pressure dependent sensitivities, are critical to acquiring concentrations of atmospheric species with minimal uncertainties. Only through calibrations can the accuracy of measurements be characterized and allow for robust comparisons with other measurements and with models to expand our current understanding of chemistry that occurs within our atmosphere.

*Author contributions.* K.H, C.E, M.M, H.H, U.J, and M.R formulated the original concept and designed the APACHE system. D.M, H.H, and U.J prototyped, developed, and characterized the APACHE system. T.K, D.M, and H.H developed and performed the CFD simulations. D.M prepared the manuscript with contributions from all coauthors.

*Acknowledgments.* We would like to take this opportunity to give a thank you to the in-house workshop at the Max Planck Institute for Chemistry for construction and guidance in the development of APACHE. Additionally, a special thank you to Dieter Scharffe for his assistance and advice during the development stage of this project.

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
