# Peer review of "Calibration of an airborne HOX instrument using the All Pressure Altitude based Calibrator for HOX Experimentation (APACHE)"

_Atmospheric Measurement Techniques, 2019_

## Referee Comment (RC1) · Anonymous Referee #1 · 30 Dec 2019

Review of Atmos. Meas. Tech. Manuscript (#amt-2019-439) "Calibration of an airborne HOx instrument using the All Pressure Altitude based Calibrator for $HO_X$ Experimentation (APACHE)" by D. Marno et al.

**General Comments**

In this work, a new airborne HOx calibration system was developed and evaluated to mimic the conditions (e.g., varying pressure, temperature, and humidity) during a typical flight. This kind of work is important to determine the response of HOx instruments for the accurate airborne measurements of OH and $HO_2$, which is the key to understand the atmospheric photochemistry. A computational fluid dynamics model (COMSOL) was used to simulate the fluid dynamics in the calibrator. Two actinometric methods based on the photolysis of ozone and N2O (used in ground-based calibrator) were used to determine the actinic flux of the mercury lamp that is used to generate OH and HO2. Overall I found this manuscript needs major revisions. The difference in actinic flux measurement using the two methods is quite large. I would suggest conducting the actinic flux measurement in APACHE using the photolysis of N2O to rule out any uncertainties in transferring the ground calibration to airborne calibration. Section 5 is particularly lean and not well organized. More details and discussion should be included in this section (see details below). I also found many errors in equations and units and tried to point most of them out. Please check out the entire manuscript. I would ask the authors to consider the following special comments in their revision.

**Special Comments**

1. L.18: For ground-based HOx instruments … (remove systems)

2. L.26: Define COMSOL at its first appearance.

3. L.47: "Other methods have also been … (Remove "However")

4. L.48: the CIMS work by C. Cantrell and L. Mauldin should also be cited here.

5. L.61-69: Start this with a new paragraph. At the end of this paragraph (or maybe start a third paragraph), you might want to mention what was done in this work (e.g. establishment and evaluation of the APACHE, etc.)

6. L.75: Define APACHE at its first appearance in the main text even though you have defined it in the abstract.

7. L.92: Figure 2 (capital F). Please check this throughout the manuscript.

8. Fig. 1: "Controlled humidity airflow of 300 sccm": is the 300 sccm of humidified air is enough to vary the humidity in the total flow of 200-900 sL/min mentioned in L.105?

9. Caption of Figure 1: Maybe change it to "Overview of the APACHE system and the pre-mixing setup. A picture at the bottom shows the perforated stainless steel plates with wool mesh."

10. L.107: The word "respectively" is used but the air speed changes by a factor of less than 2 (0.9 to 1.5 m/s) while the pressure changes by a factor of 4 (from 250 to 1000 mbar). I understand the total mass flow rate was adjusted accordingly. Please clarify this and

maybe remove respectively and say the pressure **from** 250 **to** 1000 mbar. Also because of ram effect during flight due to the installation of a choke point in the shroud (L.131) , the ambient air pressure can potentially more than 1000 mbar. Have the calibration system tested a little over 1000 mbar?

11. L.168, **where,** $W_{z1\ pwr}$ is …

12. Eq. (1) and (2): I would suggest using [OH] and [HO$_2$] for OH and HO$_2$ mixing ratios or concentrations. Please check this out for the entire manuscript. Also it seems to me that the last term ($C_{OH(2)}/C_{OH}*S_{OH}$) needs to take the laser power in the first and second axes into account (unless $Wz1_{\ power}$ and $Wz2_{\ power}$ are the same, which is unlikely) and assume there is little OH loss between the 2 axes. The OH signal in the second axis ($S_{OH(2)}$) should be:

$S_{OH(2)} = [OH]*C_{OH(2)}*Wz2_{\ power} = S_{OH}/(C_{OH}*Wz1_{\ pwr})\ *C_{OH(2)}*Wz2_{\ power}$

Please check and correct this.

13. L.179: I believe the term $Wz_{pwr}$ the should be a denominator in Eq.(4) as the units for $C_{OH}$ should be cts cm$^3$ molecule$^{-1}$ s$^{-1}$ mW$^{-1}$. Also here cm$^3$ molecule$^{-1}$ is used, while in L.170 pptv$^{-1}$ is used. Please be consistent and check this out for the entire manuscript.

14. L.189: White cell (capital W)

15. L.199: again the units in the denominator are not correct because C0 has units of cts cm$^3$ molecule$^{-1}$ s$^{-1}$ as mentioned in L.183, assuming $S_{OH}$ has units of cts s$^{-1}$.

16. L.206: see the above comment for the issue of units.

17. Caption of Figure 3: **dash-dotted** blue line and **dashed** red line.

18. L.211: Table 1 (capital T)

19. L.213: change pure to purified.

20. L.241: the units for $F_{184.9\ nm}$ should be photons **cm$^{-2}$** s$^{-1}$.

21. L.288-289: were the air flow speed profiles measured at different pressures, e.g., such as pressures lower than 920 mb to simulate conditions at high altitudes during flight?

22. L.297: Spell out COMSOL.

23. L.309-315: the disagreement could also be due to the uncertainty in the COMSOL model simulation.

24. Figure 6: the air flow speed within APACHE is really unified, even close to the wall. This is good.

25. L.316: do you mean discrete instead of discreet?

26. Caption of Figure 7: "The black arrows depict the flow direction." It is hard for me to see those arrows. Maybe include a big arrow on each plot to show the flow direction instead?

27. L.361: Please add "In Table 2" at the beginning of this sentence.

28. L.362: streamline (remove s or use streamlines in other places)

29. L.366: **F**igure 8 and **T**able 2

30. L.368: **On** the APACHE walls.

31. L.377: "between **the lamp** and a quartz wall" to be clear.

32. L.392-392: Martinez et al., 2010 is referred here, but I think at least a brief description of the ground-based calibration system should be given, especially the method to determine the actinic flux of the Hg lamp using the photolysis of N2O to provide the context for Table 3. Otherwise readers may have no idea why NO monitor/N2O cross section are suddenly mentioned in Table 3.

33. Later I found the difference of the two methods is quite large (~20%). I wonder if it is possible to conduct the actinic flux measurement in APACHE using the photolysis of N2O directly so that any uncertainties in transferring the ground calibration to airborne calibration will not affect this difference.

34. L.397: "…when the smaller 0.8 mm critical **orifice** was used."

35. L.418: Do these OH and HO$_2$ occur inside APACHE during the transport of air flow from the UV radiation zone and HORUS inlet? Please specify.

36. L.426: Duplicate definition as this has been defined in L.235.

37. L.457: units for F$_\beta$ should be photons cm$^{-2}$ s$^{-1}$.

38. L.458: Table 3 should be referred here.

39. L.459-460: Martinez et al., 2010 should be referred here.

40. Section 5: Results and Discussion: this section is very lean. Some results in Section 4 could go into this section (e.g., the results for the two methods to determine the Hg lamp actinic flux). There is also no mention how the individual measurements of overall sensitivity (1$^{st}$ row of Figure 10) are used to calculate OH and HO2 mixing ratios in the real airborne measurements. For example, the HO$_2$ sensitivity in the 2$^{nd}$ axis varied by a factor of 2 (20 vs. 10 cts/s/pptv/mW) at the internal density of 1.5E17 cm$^{-3}$. What sensitivity to use for the real measurements with internal densities between these two calibration points? Also any plan/future work to conduct more calibrations to get a better statistics and possibly to draw a smooth calibration fitted line as a function of internal pressure as shown in Figure 3?

41. L.489: Table 6 is mentioned before the appearance of Table 5.

42. L.495: "…resulting in **the transmission** for both OH and HO$_2$ to be…"

43. L.498: ".. the time it takes **for** air to flow…"

44. L.522-526: this paragraph is out of the context of this section. I would suggest moving this paragraph and some actinometric results in Section 4 to a new subsection of 5.2.

45. L.524-526: Again units for F$_\beta$ should be photons cm$^{-2}$ s$^{-1}$ or cm$^{-2}$ s$^{-1}$.

46. Again I would suggest conducting the actinic flux measurement in APACHE using the photolysis of N$_2$O directly.

47. Section 5.2. Absolute Calibration Uncertainty: this section is very lean and more discussion can be included

48. L.531: **T**ables 5 to 8.

49. Table 5: units for $F_\beta$ should be photons cm$^{-2}$ s$^{-1}$ or cm$^{-2}$ s$^{-1}$.  Also a temperature range of 282-302 K is given but no mention in the text how it was varied within APACHE.

50. Table 7: this should go Section 5.1 where transmissions are discussed.

51. Table 6 and the 3$^{rd}$ row in Figure 10: details about how the term cN* internal density is calculated/measured should be given.

52. L.559, and 562: the actinic flux of the mercury lamp should be photons cm$^{-2}$ s$^{-1}$.

53. Figure 10: the 1$^{st}$ row: the units should be cts **s$^{-1}$** pptv$^{-1}$ mW$^{-1}$.

54. Figure 10: "Row C is (C) is internal density and cN".  Do you mean "Row C is the product of internal density and cN"?  I don't understand how cN is calculated.

---

## Referee Comment (RC2) · Anonymous Referee #2 · 18 Jan 2020

In this paper Marno et al. demonstrate the first results from the "APACHE" chamber designed to calibrate and characterise the Mainz airborne "HORUS" OH and HO2 instrument. The results show the APACHE chamber operating on the ground under controlled conditions to calibrate HORUS, but it is designed also to be operated on the HALO aircraft when OH and HO2 measurements will be made, in order to calibrate in flight.

The development of a device to calibrate for OH and HO2 measurements in flight is a very difficult challenge, not only does the sensitivity of the instrument vary with a change in the pressure and temperature sampled (which changes with altitude),and

also the level of water vapour, but also the losses between the point of OH and HO2 generation in the calibrator and sampling by HORUS change also (there would be losses also for ambient OH and HO2 which are to be measured). For the former, the change in sensitivity owing to changes in parameters with altitude after the HORUS inlet can be experimentally determined via the calibration – but in this paper these are investigated through calculations also. For the latter, i.e. losses in OH from the point of generation (lamp) and the HORUS inlet need to be characterised experimentally – and understood. CFD calculations are used to simulate the flowfield within APACHE for comparison with experiment.

The description of a device to generate known concentrations of OH and HO2, and its characterisation and comparison with simulations, given the range of parameters, is complex. Likewise the sensitivity of the instrument measuring OH and HO2 and how this varies with sampling pressure is also complex – and so naturally this paper is complex and many parameters have to be explained and how they change with pressure explained. However, this is critical, as OH and HO2 are highly reactive and can be lost both in the gas-phase and at surface. The authors have made the paper fairly clear – as the characterisation is quite complex – but some further clarity is needed. The experiments appear to have been carefully performed, and many of my comments are aimed to help improving clarity for the reader.

It is not clear from the paper whether the APACHE/HORUS device has been used in flight already, as this reports experiments done in a controlled environment on the ground – and perhaps something about how it performs in flight would be useful to include, and comparison with the ground performance. The paper is an impressive piece of work – and the APACHE/HORUS is quite a feat of engineering and the thorough characterisation of APACHE and HORUS is critical to give confidence in the OH and HO2 measurements from HORUS on HALO. The paper is suitable for AMT, and the development of a calibration source for use inflight for OH measurements is very important, and a considerable achievement. There is a lot covered in this paper, but

some further details/clarifications are needed in some places. See comments below.

More specific comments.

Abstract.

A key result is that the two actinometric approaches agree fairly well, and as well as the average it would be good also to give the level of agreement also. Say what the two approaches are. What pressure is relevant for the value stated, as you say "depending on pressure", which is not clear?

Although the paper is about APACHE and its characterisation, I think readers will want to know what the sensitivity is of HORUS determined with APACHE. Could the expected C factors be stated for OH and HO2, and the derived limits of detection, and how these are predicted to vary with altitude, also be given in the abstract.

The overall accuracy of the calibration ought to be stated also in the abstract from the use of APACHE. This is given in some detail in the paper but there is nothing here. A few more numbers summarising actual performance needed in the abstract.

Also, "controlled environment" is a bit unclear, please make clear that this is on the ground, rather than results being presented of APACHE used under "a controlled environment" on the aircraft in flight.

Introduction.

46. The referencing is rather selective, please also include Juelich and Leeds LIF references (zeppelin and aircraft measurements also). For CIMS include some Eisele group references also (and subsequent including Mauldin/Cantrell which have also flown).

Figure 1. The APACHE shown here is for the controlled environment on the ground – make clear in the figure caption. Looking at Figure 2, the left hand side of APACHE would be a bit different when on the aircraft? (no inflow from mixing blocks?)

96, replace "being" with "is"

107. Is the 0.9 to 1.5 ms-1 in APACHE over the pressure range the same as the flow velocity at the same pressure when sampling on the aircraft. In line 132 the "choke" on the aircraft nacelle is used to lower the flow velocity to < 21 ms-1, but not clear if < 21 ms-1 means it will be similar to the 0.9-1.5 ms-1 as in the controlled experiments on the ground? < 21 ms-1 could cover a wide range.

124 – say also there is a critical orifice at the end of the IPI, this was not clear (and not labelled in Figure 2).

There is both a HORUS inlet, and a IPI critical orifice, and I think the presence of these two needs to be clearer. In figure 2 I suggest, that both the HORUS inlet and also the IPI critical orifice have a label. Also both "IPI orifice", "HORUS inlet" and "IPI critical orifice" are used. In line 128, is "IPI orifice" the "HORUS inlet" which samples from APACHE, or the "IPI crictical orifice" which is between the IPI and the 2 fluorescence cells? I think the former as the choke point is then mentioned which slows the flow from the aircraft speed to a slower flow in APACHE?

132. "sample velocity of HORUS", this means the flow within APACHE at which HORUS sampled perpendicularly? Is 44-53 ms-1 what is expected on the aircraft?

Figure 2. label the critical orifice in the IPI and also HORUS inlet for clarity (as discussed above).

144. As an IPI is used, it would be worth mentioning OH-WAVE (on to off resonance) and OH-CHEM, otherwise not clear of the purpose of the IPI. All the experiments performed here are OH-WAVE – presumably results of OH-CHEM in a controlled environment (to show all OH removed etc.) will be discussed in another paper. The IPI is present here but not used.

149. Again the referencing of papers is selective to a couple of groups only who use LIF.

153. Quantitative conversion is mentioned here. can a % be given, as it is not possible

to achieve 100% owing to OH+NO + M = HONO + M meaning that not all of the HO2 conversion to OH remains as OH. What is the % that is achieved here? What flow of NO is added?

180 "where" small w

202 – state the size of the critical orifice here. (diameter)

Fig 3 – make clear this is a schematic only – rather than any actual performance of the HORUS. Could point to fig 10 where this is shown. Also in the caption, the dotted blue line is for "OH transmission", whereas in the figure it is "wall loss".

219 – split – and 1 in the units

230. Juelich showed that the reaction of H* with O2 did not lead to OH, rather that 100% of H went to HO2, so worth referencing that.

Table 1. For (IV) CSTR, was the OH generated through UV irradiation of the VOC, or of another precursor? Certainly the decay rate of the VOC is used to determine the OH. Also reference Winiberg et al. 2015 (in the reference list) who used the decay of a hydrocarbon to calibrate for OH in a chamber with a LIF instrument (agreeing well with method I, water paper photolysis).

238, "where", small w

268. The exhaust from the pumps are at a different pressure when in flight compared to when the exhausts are exposed 1 atm, and this is taken account of by matching to ambient pressures in flight – that is good. Was the same pumping system used for the APACHE testing on the ground as the pumps that will be used (or are used) in flight (which might be 400 Hz pumps from the aircraft power)? (different pumps or pumps used with different motors may have different capacities).

305 "from the measured..."

Figure 6. Can it made clear what is meant by "internal wall of APACHE", perhaps by

cross-referencing to figure 1?

240. The number of sig figs in the error 179 +/-20 does not seem consistent with the sig figs quoted in the errors in brackets for the other units.

361. L, C, and R term are introduced, to make clearer, say which figure they are in – otherwise not clear what referring to.

371. How is 22.2 % loss known for OH and HO2 the inlet? (HORUS inlet). Also, one might expect the loss to be higher for the more reactive OH? Please expand a little.

Figure 8. What [H2O] the same for all the pressures? Perhaps add this value.

Tabel 2. Right hand column – OH (ppt) also?

395. The IPI critical orifice diameter is given here – but needs to be given earlier as well when this orifice is first introduced. What is the reason that the diameter of this orifice is changed from 1.4 mm to 0.8 mm for the controlled experiments on the ground?

439 and 441, another "where" to change

457 and elsewhere, for the units of flux of the light should this be "photons s-1", or even also per unit area?

Section 5 is the results, and quite a few are shown, but compared with the rest of the paper this is fairly short, and the discussion ought to be extended a little to fully exploit the results – what behaviour is therefore expected from aircraft measurements based on the lab work?

495. The losses at the inlet were the same for OH and HO2? Some further discussion of this as might expect OH to lost more.

498 "where"

Page 20 – I found this page difficult to follow, there were a lot of losses discussed, quantified by the alpha values, for various stages of the airflow, e.g. the meanings of

equations 16-18 and the discussion around this was confusing.

522. Remind reader of the two actinometric methods again (as not much detail was give on these two methods earlier).

Section 5.2 seems to be a series of tables 5-8, and a big figure, and there is virtually no text to go with this? Some further discussion is needed to bring this all together, given it is the main results from the paper. From the C factors presented , e.g. in Table 8, can the LOD of the instrument be presented, and this compared with expected levels of OH and HO2 in the atmosphere during the flights?

Figure 10. For the second row on quenching, link this to an equation used in the text – the label of the plot "Overall quenching" is unclear – and some link to the relevant part of the text is needed. Likewise for the other panels. for the first row, the y label is "Overall sensitivity" which I assume is the C(OH) factors etc., and an explicit link should be made. Likewise ALHPA (total) – refer to the equation where that is in the text.

554. The losses of HOx is discussed for the operation of APACHE during the controlled conditions ground testing. Can this be compared with the expected losses during flight when the flow velocity within APACHE may be a somewhat different (or a statement making clear the velocity within APACHE will be the same as here, or similar).

566 "is" missing after "system"

567 – experienced in flight is mentioned, but make clear again that the tests presented here are on the ground.

568. 17-18% overall uncertainty (1 sigma) – explain why this is "suitable" for a calibration approach. Mention is needed of what the measurements will be used for – to compare with OH and HO2 calculations from an atmospheric model – for which there is an uncertainty also – and a robust comparison can only be done if the measurements are accurate to a certain %, etc.

The paper focusses on pressure and water vapour, can any comments be made about

the expected change in performance (e.g. losses on surfaces, or lamp) with changes in temperature during flights?

---

## Author Comment (AC1) · 26 Feb 2020

Review of Atmos. Meas. Tech. Manuscript (#amt-2019-439) "Calibration of an airborne HOx instrument using the All Pressure Altitude based Calibrator for $HO_X$ Experimentation (APACHE)" by D. Marno et al.

We are thankful to the reviewer for the helpful and constructive comments.

**General Comments**

In this work, a new airborne HOx calibration system was developed and evaluated to mimic the conditions (e.g., varying pressure, temperature, and humidity) during a typical flight. This kind of work is important to determine the response of HOx instruments for the accurate airborne measurements of OH and $HO_2$, which is the key to understand the atmospheric photochemistry. A computational fluid dynamics model (COMSOL) was used to simulate the fluid dynamics in the calibrator. Two actinometric methods based on the photolysis of ozone and N2O (used in ground-based calibrator) were used to determine the actinic flux of the mercury lamp that is used to generate OH and HO2. Overall I found this manuscript needs major revisions. The difference in actinic flux measurement using the two methods is quite large. I would suggest conducting the actinic flux measurement in APACHE using the photolysis of N2O to rule out any uncertainties in transferring the ground calibration to airborne calibration. Section 5 is particularly lean and not well organized. More details and discussion should be included in this section (see details below). I also found many errors in equations and units and tried to point most of them out. Please check out the entire manuscript. I would ask the authors to consider the following special comments in their revision.

In light of the comments provided, we have made changes and provided clarification to the paper. Regarding the actinic flux measurements, we originally considered the lamp being a point source, which is wrong as the diameter of the lamp tube is 19mm. When considering the lamp as an respectively extended source of light with the corresponding beam profile we achieve a convergence between the two flux measurements with the HORUS transfer standard flux of 6.9 ($\pm$ 1.1) x$10^{14}$ photons cm$^{-2}$ s$^{-1}$, and the Ozone experiment yielding 6.11 ($\pm$ 0.8) x$10^{14}$ photons cm$^{-2}$ s$^{-1}$. The agreement between the two experiments have improved from a zeta score of 0.88 to 0.59, with the overall flux value being 6.37 ($\pm$ 1.3) x$10^{14}$ photons cm$^{-2}$ s$^{-1}$. This value is calculated by taking the average of the two methods weighted by their uncertainties. Section 5 has been merged with section 4, as we feel that the whole of section 4 entails results and discussion. Discussed elements that were in section 5 have been organized and expanded upon, please see comments and revised document.

**Special Comments**

1. L.18: For ground-based HOx instruments … (remove systems)

   Deleted "Systems"

2. L.26: Define COMSOL at its first appearance.

3. We have described  COMSOL as a computational fluid dynamics model. Its origin is FEMLAB, a former toolbox of Matlab, which name is derived from 'Finite-Element-Method-Laboratory'.

4. L.47: "Other methods have also been … (Remove "However")

   Removed "However"

5. L.48: the CIMS work by C. Cantrell and L. Mauldin should also be cited here.

   "Cited C. Cantrell and L. Mauldin" .

6. L.61-69: Start this with a new paragraph.  At the end of this paragraph (or maybe start a third paragraph), you might want to mention what was done in this work (e.g. establishment and evaluation of the APACHE, etc.)

   Now is a separate paragraph. Containing what was done in this work.

7. L.75: Define APACHE at its first appearance in the main text even though you have defined it in the abstract.

   Defined APACHE in its first appearance in the main text.

8. L.92: Figure 2 (capital F).  Please check this throughout the manuscript.

   All references to a figure or table in text or otherwise have been capitalized as Figure or Table.

9. Fig. 1: "Controlled humidity airflow of 300 sccm": is the 300 sccm of humidified air is enough to vary the humidity in the total flow of 200-900 sL/min mentioned in L.105?

   Typo. It was 300 sL min$^{-1}$. Figure corrected.

10. Caption of Figure 1: Maybe change it to "Overview of the APACHE system and the premixing setup.  A picture at the bottom shows the perforated stainless steel plates with wool mesh."

    Changed caption for Figure 1.

11. L.107: The word "respectively" is used but the air speed changes by a factor of less than 2 (0.9 to 1.5 m/s) while the pressure changes by a factor of 4 (from 250 to 1000 mbar). I understand the total mass flow rate was adjusted accordingly. Please clarify this and

    maybe remove respectively and say the pressure **from** 250 **to** 1000 mbar.  Also because of ram effect during flight due to the installation of a choke point in the shroud (L.131) , the ambient air pressure can potentially more than 1000 mbar.  Have the calibration system tested a little over 1000 mbar?

    During testing it was found that APACHE is capable of operating at pressures exceeding 1000 mbar.  However, the main focus of this study was to investigate APACHE operation to calibrate the HORUS instrument for the HALO (High Altitude Long Range) aircraft at altitudes above the boundary layer. Only below 1.5km the pressure is due to the ram pressure larger than 1000 mbar

12. L.168, **where,** $W_{z1\ pwr}$ is …

Added "Where,"

13. Eq. (1) and (2): I would suggest using [OH] and [HO$_2$] for OH and HO$_2$ mixing ratios or concentrations. Please check this out for the entire manuscript. Also it seems to me that the last term ($C_{OH(2)}$/$C_{OH}$*$S_{OH}$) needs to take the laser power in the first and second axes into account (unless Wz1 $_{power}$ and Wz2 $_{power}$ are the same, which is unlikely) and assume there is little OH loss between the 2 axes. The OH signal in the second axis ($S_{OH(2)}$) should be:

$S_{OH(2)}$ = [OH]*$C_{OH(2)}$*Wz2 $_{power}$ = $S_{OH}$/($C_{OH}$*Wz1 $_{pwr}$) *$C_{OH(2)}$*Wz2 $_{power}$

Please check and correct this.

Checked and agree with the proposed changes.

14. L.179: I believe the term $Wz_{pwr}$ the should be a denominator in Eq.(4) as the units for $C_{OH}$ should be cts cm$^3$ molecule$^{-1}$ s$^{-1}$ mW$^{-1}$. Also here cm$^3$ molecule$^{-1}$ is used, while in L.170 pptv$^{-1}$ is used. Please be consistent and check this out for the entire manuscript.

Eq. (4) has been adapted with the consideration of the following:

The c0 coefficient actually has the units (cts pptv$^{-1}$ s$^{-2}$ cm$^3$ molecule$^{-1}$ mW$^{-1}$) i.e in calibrations it is normalized by laser power, Boltzmann correction, quenching (s), internal density (molecules cm$^{-3}$). During flight and c0 is multiplied by Boltzmann correction, quenching (s), internal density (molecules cm$^{-3}$) resulting in the sensitivity $C_{OH}$ having the units cts s$^{-1}$ pptv$^{-1}$ mW$^{-1}$. $C_{OH}$ is then scaled by the actual power measured in flight resulting in the units, cts s$^{-1}$ pptv$^{-1}$. Then the averaged 5 Hz measured signal (averaged to cts in a second) during flight (see Eq. 1) is subsequently divided by the laser power scaled $C_{OH}$, resulting in the units pptv for OH.

15. L.189: White cell (capital W)

Capitalized W.

16. L.199: again the units in the denominator are not correct because C0 has units of cts cm$^3$ molecule$^{-1}$ s$^{-1}$ as mentioned in L.183, assuming $S_{OH}$ has units of cts s$^{-1}$.

See comment 13. Units for c0 was not correctly described.

17. L.206: see the above comment for the issue of units.

See comment 13.

18. Caption of Figure 3: **dash-dotted** blue line and **dashed** red line.

Corrected.

19. L.211: Table 1 (capital T)

Capitalized the T.

20. L.213: change pure to purified.

Pure in this statement means that only synthetic air is used to calibrate with. I.e. no other type of gas is used as a medium. We have changed "pure" to "only" to emphasize that only syn air is used as the medium for calibration.

21. L.241: the units for $F_{184.9\ nm}$ should be photons **cm$^{-2}$** s$^{-1}$.

Units corrected to photons cm$^{-2}$ s$^{-1}$.

22. L.288-289: were the air flow speed profiles measured at different pressures, e.g., such as pressures lower than 920 mb to simulate conditions at high altitudes during flight?

With pressures below 920 mbar, the reading from the differential pressure sensor was close to or below its resolution. Hence the need to utilize other methods to parameterize the flow conditions within APACHE.

23. L.297: Spell out COMSOL.

Defined COMSOL again as a computational fluid dynamics model (CFD).

24. L.309-315: the disagreement could also be due to the uncertainty in the COMSOL model simulation.

Added a comment stating this.

25. Figure 6: the air flow speed within APACHE is really unified, even close to the wall. This is good.

26. L.316: do you mean discrete instead of discreet?

Yes, this is a misspelling.

27. Caption of Figure 7: "The black arrows depict the flow direction." It is hard for me to see those arrows. Maybe include a big arrow on each plot to show the flow direction instead?

We have increased the arrow size to make them clearer.

28. L.361: Please add "In Table 2" at the beginning of this sentence.

Added Table 2

29. L.362: streamline (remove s or use streamlines in other places)

By this, we mean literally on the leftmost streamline for L or rightmost streamline for R, C is in the middle of the streamlines. We have checked that such plural or singular usage is consistent.

30. L.366: **F**igure 8 and **T**able 2

Capitalized

31. L.368: **On** the APACHE walls.

Changed "at" to "on".

32. L.377: "between **the lamp** and a quartz wall" to be clear.

Agreed and applied the change.

33. L.392-392: Martinez et al., 2010 is referred here, but I think at least a brief description of the ground-based calibration system should be given, especially the method to determine the actinic flux of the Hg lamp using the photolysis of N2O to provide the context for Table 3. Otherwise readers may have no idea why NO monitor/N2O cross section are suddenly mentioned in Table 3.

We have included a short description and equation with reference to Martinez et al., 2010, showing where the NO monitor and NO standard terms in table 3 are coming from.

34. Later I found the difference of the two methods is quite large (~20%). I wonder if it is possible to conduct the actinic flux measurement in APACHE using the photolysis of N2O directly so that any uncertainties in transferring the ground calibration to airborne calibration will not affect this difference.

The difference in the original flux values may appear large however, given their uncertainties the zeta score was 0.88, suggesting agreement within the combined uncertainty of both measurements. However, in light of the comments and suggestions, the calibrations and terms therein have be checked, reevaluated and adjusted when considering the lamp as an extended source of light with a corresponding beam profile. By doing this, the two methods converge, 6.9 ($\pm$ 1.1) x$10^{14}$ photons cm$^{-2}$ s$^{-1}$ for method A, and 6.11 ($\pm$ 0.8) x$10^{14}$ photons cm$^{-2}$ s$^{-1}$ for method B. The agreement between the two experiments has a zeta score of 0.59, meaning they agree to within 59 % of one sigma of their combined uncertainties, suggesting agreement.

35. L.397: "…when the smaller 0.8 mm critical **orifice** was used."

Added "orifice".

36. L.418: Do these OH and HO$_2$ occur inside APACHE during the transport of air flow from the UV radiation zone and HORUS inlet? Please specify.

Yes, we calculate this based on the recommended reaction rates, number densities occurring in APACHE, and the calculated transit times that occur in APACHE between the lamp and the HORUS inlet.

37. L.426: Duplicate definition as this has been defined in L.235.

The equation is used again here as the discussion is building up from it. We believe it is easier for the reader to follow the discussion if they do not have to flip back several pages to check what is being referred to in this section of the discussion.

38. L.457: units for F$_\beta$ should be photons cm$^{-2}$ s$^{-1}$.

Corrected the units

39. L.458: Table 3 should be referred here.

Table 3 in now referred here.

40. L.459-460: Martinez et al., 2010 should be referred here.

Martinez et al., 2010 in now referred here.

41. Section 5: Results and Discussion: this section is very lean. Some results in Section 4 could go into this section (e.g., the results for the two methods to determine the Hg lamp actinic flux). There is also no mention how the individual measurements of overall sensitivity (1$^{st}$ row of Figure 10) are used to calculate OH and HO2 mixing ratios in the real airborne measurements. For example, the HO$_2$ sensitivity in the 2$^{nd}$ axis varied by a factor of 2 (20 vs. 10 cts/s/pptv/mW) at the internal density of 1.5E17 cm$^{-3}$. What sensitivity to use for the real measurements with internal densities between these two

calibration points?  Also any plan/future work to conduct more calibrations to get a better statistics and possibly to draw a smooth calibration fitted line as a function of internal pressure as shown in Figure 3?

Section 5 is now merged with section 4, with some aspects expanded upon. We have included a description and equation showing how the c0, c1,c2 (otherwise labelled as a grouped term cN in the figure) are calculated.

In row A Figure 10 we have decided to provide a smoothed calibration curve much like the one shown in figure 3. Once c0, c1,c2 are known, and quenching, internal density and transmission efficiency are quantified with consideration of the measured internal temperatures and pressures, one can use equation 4, to adequately resolve the sensitivity within 2 sigma of the uncertainties.  These points have been added into section 4.3.1. Additionally we have included a paragraph clarifying how these terms are then used to quantify the sensitivity for airborne measurements. Figure 11 has been included to show how sensitivity, $HO_X$ transmission and detection limits look like when quantified using measured temperatures and pressures values in HORUS under flight  conditions .

42. L.489: Table 6 is mentioned before the appearance of Table 5.

Table 5 is now mentioned before mentioning Table 6.

43. L.495: "…resulting in **the transmission** for both OH and $HO_2$ to be…"

Added "the transmission".

44. L.498: ".. the time it takes **for** air to flow…"

Added "for".

45. L.522-526: this paragraph is out of the context of this section.  I would suggest moving this paragraph and some actinometric results in Section 4 to a new subsection of 5.2.

Paragraph moved to calibration uncertainty section, 4.3.2. Where a fuller discussion regarding uncertainty is present.

46. L.524-526: Again units for $F_\beta$ should be photons $cm^{-2}$ $s^{-1}$ or $cm^{-2}$ $s^{-1}$.

Changed to units for $F_\beta$ to photons $cm^{-2}$ $s^{-1}$ or $cm^{-2}$ $s^{-1}$.

47. Again I would suggest conducting the actinic flux measurement in APACHE using the photolysis of $N_2O$ directly.

Addressed in previous sections, and opening statement

48. Section 5.2. Absolute Calibration Uncertainty: this section is very lean and more discussion can be included

This section has been incorporated into the Evaluation of instrumental sensitivity section. In hindsight, we believe that it is clearer for the reader to follow the discussion and to realize where the uncertainties are sourced from and to what scale they impact the final sensitivity values.

49. L.531: **T**ables 5 to 8.

    Capitalized T

50. Table 5: units for $F_\beta$ should be photons cm$^{-2}$ s$^{-1}$ or cm$^{-2}$ s$^{-1}$. Also a temperature range of 282-302 K is given but no mention in the text how it was varied within APACHE.

    Temperature ranges now discussed in section 2.2

51. Table 7: this should go Section 5.1 where transmissions are discussed.

    See comment 47.

52. Table 6 and the 3$^{rd}$ row in Figure 10: details about how the term cN* internal density is calculated/measured should be given.

    Included equation and discussion regarding how cN is calculated se Eq 14.

53. L.559, and 562: the actinic flux of the mercury lamp should be photons cm$^{-2}$ s$^{-1}$.

    Corrected the units

54. Figure 10: the 1$^{st}$ row: the units should be cts **s$^{-1}$** pptv$^{-1}$ mW$^{-1}$.

    Corrected the units

55. Figure 10: "Row C is (C) is internal density and cN". Do you mean "Row C is the product of internal density and cN"? I don't understand how cN is calculated.

    Included equation and discussion regarding how cN is calculated see Eq. 14,15 and 16.

---

## Author Comment (AC2) · 26 Feb 2020

In this paper Marno et al. demonstrate the first results from the "APACHE" chamber designed to calibrate and characterise the Mainz airborne "HORUS" OH and HO2 instrument. The results show the APACHE chamber operating on the ground under controlled conditions to calibrate HORUS, but it is designed also to be operated on the HALO aircraft when OH and HO2 measurements will be made, in order to calibrate in flight.

The development of a device to calibrate for OH and HO2 measurements in flight is a very difficult challenge, not only does the sensitivity of the instrument vary with a change in the pressure and temperature sampled (which changes with altitude),and also the level of water vapour, but also the losses between the point of OH and HO2 generation in the calibrator and sampling by HORUS change also (there would be losses also for ambient OH and HO2 which are to be measured). For the former, the change in sensitivity owing to changes in parameters with altitude after the HORUS inlet can be experimentally determined via the calibration – but in this paper these are investigated through calculations also. For the latter, i.e. losses in OH from the point of generation (lamp) and the HORUS inlet need to be characterised experimentally – and understood. CFD calculations are used to simulate the flowfield within APACHE for comparison with experiment.

The description of a device to generate known concentrations of OH and HO2, and its characterisation and comparison with simulations, given the range of parameters, is complex. Likewise the sensitivity of the instrument measuring OH and HO2 and how this varies with sampling pressure is also complex – and so naturally this paper is complex and many parameters have to be explained and how they change with pressure explained. However, this is critical, as OH and HO2 are highly reactive and can be lost both in the gas-phase and at surface. The authors have made the paper fairly clear – as the characterisation is quite complex – but some further clarity is needed. The experiments appear to have been carefully performed, and many of my comments are aimed to help improving clarity for the reader.

It is not clear from the paper whether the APACHE/HORUS device has been used in flight already, as this reports experiments done in a controlled environment on the ground – and perhaps something about how it performs in flight would be useful to include, and comparison with the ground performance. The paper is an impressive piece of work – and the APACHE/HORUS is quite a feat of engineering and the thorough characterisation of APACHE and HORUS is critical to give confidence in the OH and HO2 measurements from HORUS on HALO. The paper is suitable for AMT, and the development of a calibration source for use inflight for OH measurements is very important, and a considerable achievement. There is a lot covered in this paper, but

some further details/clarifications are needed in some places. See comments below.

More specific comments.

Abstract.

A key result is that the two actinometric approaches agree fairly well, and as well as the average it would be good also to give the level of agreement also. Say what the two approaches are. What pressure is relevant for the value stated, as you say "depending on pressure", which is not clear?

Stated what the two approaches are, their values, and agreement in the form of zeta score. We have removed the mention of pressure as the actinic flux of the lamp is not pressure dependent, this information is discussed at greater length in the text. Not relevant here in the abstract.

Although the paper is about APACHE and its characterisation, I think readers will want to know what the sensitivity is of HORUS determined with APACHE. Could the expected C factors be stated for OH and HO2, and the derived limits of detection, and how these are predicted to vary with altitude, also be given in the abstract.

The overall accuracy of the calibration ought to be stated also in the abstract from the use of APACHE. This is given in some detail in the paper but there is nothing here. A few more numbers summarising actual performance needed in the abstract.

We have included sensitivity values and the calibration accuracy. Regarding the limits of detection we have included a figure and discussion at the end of the paper, describing how they changing during flight.

Also, "controlled environment" is a bit unclear, please make clear that this is on the ground, rather than results being presented of APACHE used under "a controlled environment" on the aircraft in flight.

We have stated here that calibrations with APACHE were performed in the lab.

Introduction.

46. The referencing is rather selective, please also include Juelich and Leeds LIF references (zeppelin and aircraft measurements also). For CIMS include some Eisele group references also (and subsequent including Mauldin/Cantrell which have also flown).

Included these references

Figure 1. The APACHE shown here is for the controlled environment on the ground – make clear in the figure caption. Looking at Figure 2, the left hand side of APACHE would be a bit different when on the aircraft? (no inflow from mixing blocks?)

We do not characterize the inlet shroud, but the HORUS instrument starting at the inlet(IPI Nozzle). In APACHE we provide a homogenous flow profile with a characterized OH profile to HORUS.

96, replace "being" with "is"

Replaced "being" with "is".

107. Is the 0.9 to 1.5 ms-1 in APACHE over the pressure range the same as the flow velocity at the same pressure when sampling on the aircraft. In line 132 the "choke" on the aircraft nacelle is used to lower the flow velocity to < 21 ms-1, but not clear if < 21 ms-1 means it will be similar to the 0.9-1.5 ms-1 as in the controlled experiments on the ground? < 21 ms-1 could cover a wide range.

We have looked into periods during take-off and landing where there are large changes in flow speed (1 to 12 m s$^{-1}$) within the shroud and we find no change in our signal that is attributable to flow speed changes across the IPI nozzle. Therefore, there is no uncharacterized loss, at a detectable level, occurring at the IPI Nozzle when flow speeds are 0.9-1.5 m s$^{-1}$ in APACHE when compared to  21 m s$^{-1}$ during flight.

124 – say also there is a critical orifice at the end of the IPI, this was not clear (and not labelled in Figure 2).

Improved the labelling in figure 2.

There is both a HORUS inlet, and a IPI critical orifice, and I think the presence of these two needs to be clearer. In figure 2 I suggest, that both the HORUS inlet and also the IPI critical orifice have a label. Also both "IPI orifice", "HORUS inlet" and "IPI critical orifice" are used. In line 128, is "IPI orifice" the "HORUS inlet" which samples from APACHE, or the "IPI crictical orifice" which is between the IPI and the 2 fluorescence cells? I think the former as the choke point is then mentioned which slows the flow from the aircraft speed to a slower flow in APACHE?

The IPI nozzle/ inlet is not a critical orifice. The critical orifice sits between IPI itself and the first detection cell. The choke point is at the end of the inner inlet shroud. We have changed the labelling throughout the paper to ensure consistency.

132. "sample velocity of HORUS", this means the flow within APACHE at which HORUS sampled perpendicularly? Is 44-53 ms-1 what is expected on the aircraft? Figure 2. label the critical orifice in the IPI and also HORUS inlet for clarity (as discussed above).

This means the sample flow speed within IPI ranged between 44-53 ms-1 during flight. We have also include at the end of this paragraph a statement explaining that the location of the critical orifice allows HORUS to sample (~ 3 - 17 sL min$^{-1}$) from the central flow that is moving through IPI (~ 51 - 230 sL min$^{-1}$) . The excess flow is removed via a perforated ring that surrounds the base of the critical orifice cone evacuated by a blower. All discussion in section 2.3 is regarding parameters and occurrences during flight. We have also included "during flight" statements here to emphasize we are talking about processes happening in and around HORUS when airborne. We have also adjusted the labelling in Figure 2 (see above).

144. As an IPI is used, it would be worth mentioning OH-WAVE (on to off resonance) and OH-CHEM, otherwise not clear of the purpose of the IPI. All the experiments performed here are OH-WAVE – presumably results of OH-CHEM in a controlled environment (to show all OH removed etc.) will be discussed in another paper. The IPI is present here but not used.

We have included the OH-WAVE and OH-CHEM discussion here. IPI was operated during calibrations as it would have been during flight as it does impact the overall sensitivity. But, as you have indicated, the inflight performance of IPI with regards to scavenging efficiency and OH-CHEM in flight will be discussed in a different publication. For this paper OH-CHEM is not the focus.

149. Again the referencing of papers is selective to a couple of groups only who use LIF.

We have included primary references that discuss directly the OH absorption spectrum. We have left the LIF references because at this point we are only discussing HORUS as a LIF instrument.

153. Quantitative conversion is mentioned here. can a % be given, as it is not possible

to achieve 100% owing to OH+NO + M = HONO + M meaning that not all of the HO2 conversion to OH remains as OH. What is the % that is achieved here? What flow of NO is added?

We have calculated the internal HONO formation in our instrument using the caaba/mecca box model initializing with HO2 and NO at the corresponding low pressure conditions experienced in flight. Note that here any reference to [OH] is in regards to OH formed from the reaction of HO2 with NO. The following figure shows the fractional HONO concentrations formed compared to the formed OH concentrations at different flight altitudes  i.e. [HONO] from the reaction $k_{OH+NO+M}$[NO][OH][M] divided by [OH] from $k_{HO2+NO}$[HO$_2$][NO]. This is to show at flight altitudes of 14 km, 9 km, 8 km and 3.5 km what percentage of OH formed from HO$_2$ + NO undergoes further reaction forming HONO internally within HORUS. The black dotted-dashed line is the maximum NO concentration (1.04 x10$^{14}$ molecules cm$^{-3}$) injected into HORUS when we are performing in-flight NO titrations. The blue dotted-dashed line shows the maximum NO concentration (0.79 x10$^{14}$ molecules cm$^{-3}$) injected into HORUS when performing normal measurements, the red dotted-dashed line shows the minimum NO concentration (6.61 x10$^{12}$ molecules cm$^{-3}$) injected into HORUS. When measuring we toggle our NO injection between these two

concentrations to resolve for $RO_2$ interference. At the low NO mode any RO2 interference in the signal is heavily suppressed as there is not sufficient NO present in HORUS to promote production of OH via RO2+NO. The higher NO addition has a better signal to noise ratio, however contains a more significant RO2 contribution. To resolve for this we perform NO titrations to resolve our HO2 conversion efficiency at every pressure level and NO concentrations being injected into HORUS. If the high NO injection signal (once corrected for conversion efficiency) is significantly higher (i.e. consistently above by more than the detection limit) than the low NO concentration signal (once corrected for conversion efficiency) we use the signal from the low NO injection mode for atmospheric HO2 measurements. If the high NO injection is not greater than the low NO injection mode ( i.e. higher by more than the detection limit of HORUS) we use the high NO injection mode as the signal to noise ratio is better.

When titrating to maximum NO concentrations, 3.3 % of formed OH is converted into HONO at 14 km, 5.4 % of formed OH is converted into HONO at 9 km, 7.8 % of formed OH is converted into HONO at 8 km, and 13.8 % of formed OH is converted into HONO at 3.5 km. These values are the upper limit of HONO formation, as the calculations assume perfect mixing of NO. Additionally in this figure for all altitudes, the low NO injection measurement mode results in less than 0.5 % of the formed OH being lost via HONO formation, which further limits the influence of HONO formation on the HO2 signal.

[Figure]

We have also determined what NO concentrations are required to cause HONO formation to have a detectable influence on the HO2 signal in HORUS, i.e. at what NO concentration does the drop in [OH] (caused by HONO formation) from the maximum titrated OH concentration value exceed the detection limit of the instrument. The table below shows these values:

| Altitude (km) | Required NO concentration in HORUS (x$10^{14}$ molecules cm$^{-3}$), to cause HONO formation to have a detectable influence |
|---|---|
| 14 | 2.36 |
| 9 | 1.52 |
| 8 | 1.14 |
| 3.5 | 0.82 |

Note: The NO concentration values quoted here are the lower limit, as these are calculated under the assumption of perfect NO mixing, and taking the minimum characterized detection limit at each altitude level.

This table shows that given the strong pressure dependence of the termolecular reaction that forms HONO, significantly higher NO concentrations (>14% than the maximum titrated concentration) are required to result in a detectable influence on the $HO_2$ signal via HONO formation. Only at flight altitudes 3.5 km and below can HONO formation have a detectable influence. However, this is only in the cases when we are titrating at these low altitudes which is not the main focus of the OMO-ASIA 2015 campaign in which HORUS took part and this study, where the main focus was and is on altitudes exceeding 8 k km. Even in the high HO2 conversion mode, applying NO in the order of 0.79 x$10^{14}$ molecules cm$^{-3}$, the HONO formation still falls below this lower limit.

This discussion regarding HONO forms part of a later publication where instrument performance (e.g. OH-CHEM and $RO_2$ interferences etc) is the focus. Alongside intercomparison with the LIF instrument from Jülich. As the too flew on HALO during the same OMO-ASIA 2015 campaign.

180 "where" small w

Changed to lower case w.

202 – state the size of the critical orifice here. (diameter)

Stated the diameter. 1.4 mm.

Fig 3 – make clear this is a schematic only – rather than any actual performance of the HORUS. Could point to fig 10 where this is shown. Also in the caption, the dotted blue line is for "OH transmission", whereas in the figure it is "wall loss".

Corrected the labelling in the figure. Explicitly describe it as a schematic.

219 – split – and 1 in the units

Corrected.

230. Juelich showed that the reaction of H* with O2 did not lead to OH, rather that 100% of H went to HO2, so worth referencing that.

Added Jülich reference.

Table 1. For (IV) CSTR, was the OH generated through UV irradiation of the VOC, or of another precursor? Certainly the decay rate of the VOC is used to determine the OH. Also reference Winiberg et al. 2015 (in the reference list) who used the decay of a hydrocarbon to calibrate for OH in a chamber with a LIF instrument (agreeing well with method I, water paper photolysis).

We have altered the description to match how they are described in the referenced publications. Added Winiberg et al., 2015 to the reference list, including what hydrocarbons were used in that study.

238, "where", small w

Changed to lower case w.

268. The exhaust from the pumps are at a different pressure when in flight compared to when the exhausts are exposed 1 atm, and this is taken account of by matching to ambient pressures in flight – that is good. Was the same pumping system used for the APACHE testing on the ground as the pumps that will be used (or are used) in flight (which might be 400 Hz pumps from the aircraft power)? (different pumps or pumps used with different motors may have different capacities).

Clarified here that the pumps used during calibrations with APACHE are the same ones that were installed on the aircraft. Also that we used a 3 phase mission power supply unit that provides the same power as on the aircraft.

305 "from the measured..."

Corrected

Figure 6. Can it made clear what is meant by "internal wall of APACHE", perhaps by cross-referencing to figure 1?

We added a small caption in figure 6 showing what we mean by Internal wall of APACHE

240. The number of sig figs in the error 179 +/-20 does not seem consistent with the sig figs quoted in the errors in brackets for the other units.

Changed the sig figs, so that they match.

361. L, C, and R term are introduced, to make clearer, say which figure they are in – otherwise not clear what referring to.

Clarified in what sense to the L, C and R terms relate to, i.e the streamlines created by the HORUS sample flow in figure 7 and 8.

371. How is 22.2 % loss known for OH and HO2 the inlet? (HORUS inlet). Also, one might expect the loss to be higher for the more reactive OH? Please expand a little.

We have adapted our discussion regarding this variable. According to the model irrespective of pressure the IPI nozzle is 22 %, suggesting that this loss is pressure independent. This value is not utilized any further. The true/characterized/measurable pressure independent loss is now characterized within the pressure independent sensitivity coefficients, which do differ between OH and HO2 at the second axis.

Figure 8. What [H2O] the same for all the pressures? Perhaps add this value.

We have included the water mixing ratio. It was kept constant at 3.2 mmol/mol.

Tabel 2. Right hand column – OH (ppt) also?

Yes pptv. Units added to this column.

395. The IPI critical orifice diameter is given here – but needs to be given earlier as well when this orifice is first introduced. What is the reason that the diameter of this orifice is changed from 1.4 mm to 0.8 mm for the controlled experiments on the ground?

This adaptation was done to enable use to relate the flux of a pre-calibrated penray lamp used on the ground based calibration device to $F_\beta$ entering APACHE

Adapted and expanded upon the reasoning:

"Since the pre-characterized ground based calibration device is designed to supply only 50 sL min$^{-1}$, and the sensitivity of airborne HORUS instrument is optimized for high altitude flying, the critical orifice diameter in HORUS was changed from the airborne configuration of 1.4 mm to a 0.8 mm on-ground* configuration. Additionally, the IPI system was switched to passive (i.e. the exhaust line to the IPI blower was capped using a kf 40 flange). This was to adapt HORUS to a mass flow that the ground based calibration device is able to provide and reduces the internal pressure within HORUS (from 18 mbar to 3.5 mbar) to optimize the sensitivity towards OH at ambient ground level pressures (~1000 mbar). The asterisk discerns terms that were quantified when the smaller 0.8 mm critical orifice was used. The calculated instrument on-ground* sensitivity was then used to translate OH and HO$_2$ concentrations produced by the uv-technik Hg ring lamp into a value for $F_\beta$."

439 and 441, another "where" to change

Changed to lower case w

457 and elsewhere, for the units of flux of the light should this be "photons s-1", or even also per unit area?

All flux units have be corrected "photons cm$^{-2}$ s$^{-1}$).

Section 5 is the results, and quite a few are shown, but compared with the rest of the paper this is fairly short, and the discussion ought to be extended a little to fully exploit the results – what behaviour is therefore expected from aircraft measurements based on the lab work?

We have expanded section 5 into section 4. We have also provided additional context and discussion in the section, including instrument behavior during a typical flight.

495. The losses at the inlet were the same for OH and HO2? Some further discussion of this as might expect OH to lost more.

See response 371.

498 "where"

Page 20 – I found this page difficult to follow, there were a lot of losses discussed, quantified by the alpha values, for various stages of the airflow, e.g. the meanings of equations 16-18 and the discussion around this was confusing.

We have expanded on points here and explicitly stated which alpha term is which and how they are summed to together to acquire the total OH and HO2 pressure dependent transmission terms.

522. Remind reader of the two actinometric methods again (as not much detail was give on these two methods earlier).

Removed this paragraph as it did not sit well within the context of the discussion at this point. We talk about the two actinometric methods again within the conclusions.

Section 5.2 seems to be a series of tables 5-8, and a big figure, and there is virtually no text to go with this? Some further discussion is needed to bring this all together, given it is the main results from the paper. From the C factors presented , e.g. in Table 8, can the LOD of the instrument be presented, and this compared with expected levels of OH and HO2 in the atmosphere during the flights?

We have expanded section 5 into section 4. We have also provided additional context and discussion in the section, including instrument behavior during flight. Including LOD.

Figure 10. For the second row on quenching, link this to an equation used in the text – the label of the plot "Overall quenching" is unclear – and some link to the relevant part of the text is needed. Likewise for the other panels. for the first row, the y label is "Overall sensitivity" which I assume is the C(OH) factors etc., and an explicit link should be made. Likewise ALHPA (total) – refer to the equation where that is in the text.

We have included an equation explaining how the quenching is calculated. Within the text and figure explicit links have been included regarding quenching, C(OH), and ALPHA$_{Total}$.

554. The losses of HOx is discussed for the operation of APACHE during the controlled conditions ground testing. Can this be compared with the expected losses during flight when the flow velocity within APACHE may be a somewhat different (or a statement making clear the velocity within APACHE will be the same as here, or similar).

We have included a figure for inflight conditions to allow for discussion and direct comparison, between controlled ground testing and in-flight losses.

566 "is" missing after "system"

Corrected the statement to "However, in this study, the APACHE calibration system has demonstrated that, within the lab, it is sufficiently capable of calibrating the airborne HORUS instrument across the pressure ranges the instrument had experienced in-flight during the OMO-ASIA 2015 airborne campaign."

567 – experienced in flight is mentioned, but make clear again that the tests presented here are on the ground.

See comment above

568. 17-18% overall uncertainty (1 sigma) – explain why this is "suitable" for a calibration approach. Mention is needed of what the measurements will be used for – to compare with OH and HO2 calculations from an atmospheric model – for which there is an uncertainty also – and a robust comparison can only be done if the measurements are accurate to a certain %, etc.

The overall uncertainty is now 22.1 – 22.6 % (1 sigma). We have adjusted this statement to be a direct comparison to the other calibration methods shown in table 1. As of this study we are not addressing an overarching scientific question, and therefore making no statement regarding the "suitability" of this uncertainty.

"The overall uncertainty of 22.1 – 22.6 % (1σ) demonstrates that this calibration approach with APACHE compares well with other calibration methods described earlier in Table 1. Accurate calibrations of instruments, particularly airborne instruments that have strong pressure dependent sensitivities, are critical to acquiring concentrations of atmospheric species with minimal uncertainties. Only through calibrations can the accuracy of measurements be characterized and allow for robust comparisons with other measurements and with models to expand our current understanding of chemistry that occurs within our atmosphere."

The paper focusses on pressure and water vapour, can any comments be made about the expected change in performance (e.g. losses on surfaces, or lamp) with changes in temperature during flights?

The APACHE system is an on ground setup, built to replicate conditions in flight . It is not installed on the aircraft. However, we have highlighted future developments of APACHE to adapt it for temperature control as well as pressure control.